# Biallelic *ADPRHL2* mutations in complex neuropathy affect ADP ribosylation and DNA damage response

Danique Beijer[1,2], Thomas Agnew[3], Johannes Gregor Matthias Rack[3], Evgeniia Prokhorova[3], Tine Deconinck[1,2], Berten Ceulemans[4], Stojan Peric[5], Vedrana Milic Rasic[6], Peter De Jonghe[1,2,7], Ivan Ahel[3], Jonathan Baets[1,2,7]

**ADP ribosylation is a reversible posttranslational modification mediated by poly(ADP-ribose)transferases (e.g., PARP1) and (ADP-ribosyl)hydrolases (e.g., ARH3 and PARG), ensuring synthesis and removal of mono-ADP-ribose or poly-ADP-ribose chains on protein substrates. Dysregulation of ADP ribosylation signaling has been associated with several neurodegenerative diseases, including Parkinson's disease, amyotrophic lateral sclerosis, and Huntington's disease. Recessive *ADPRHL2*/ARH3 mutations are described to cause a stress-induced epileptic ataxia syndrome with developmental delay and axonal neuropathy (CONDSIAS). Here, we present two families with a neuropathy predominant disorder and homozygous mutations in *ADPRHL2*. We characterized a novel C26F mutation, demonstrating protein instability and reduced protein function. Characterization of the recurrent V335G mutant demonstrated mild loss of expression with retained enzymatic activity. Although the V335G mutation retains its mitochondrial localization, it has altered cytosolic/nuclear localization. This minimally affects basal ADP ribosylation but results in elevated nuclear ADP ribosylation during stress, demonstrating the vital role of ADP ribosylation reversal by ARH3 in DNA damage control.**

## Introduction

ADP ribosylation (ADPr) is a reversible posttranslational protein modification involved in the regulation of many cellular processes including cell differentiation, metabolism, apoptosis, antiviral responses, and DNA damage repair (Gupte et al, 2017; Kim et al, 2020; Crawford et al, 2021). ADPr is synthesized by (ADP-ribosyl) transferases such as the poly(ADP-ribose)polymerase (PARP) class of enzymes, most prominently PARP1. PARP1 catalyzes the attachment of a single ADP-ribose unit (mono[ADP-ribosy]lation; MAR) from $NAD^+$ onto a target site before extending this modification into ADP-ribose polymers (poly[ADP-ribosyl]ation; PAR) (Pascal, 2018). Proteins can be ADP-ribosylated on several different amino acids, but recently it has been shown that serine residues are the most commonly modified residues in vivo (Leidecker et al, 2016; Palazzo et al, 2018; Hendriks et al, 2019; Suskiewicz et al, 2020a). Serine MAR is synthesized by the DNA repair PARPs PARP1 and PARP2 in complex with HPF1, after which MAR is extended into chains of PARylation (Gibbs-Seymour et al, 2016; Suskiewicz et al, 2020b). After DNA damage, PARP1/2 binds to DNA damage sites, modifying itself and nearby histones with MAR and PAR chains (Bonfiglio et al, 2017; Palazzo et al, 2018; Hou et al, 2019). PARP1/2-dependent ADP-ribose at DNA damage sites facilitates the recruitment and subsequent modification of DNA damage repair proteins with PAR-binding motifs or PAR-binding domains such as PAR-binding zinc finger (PBZ) domains or macrodomains (Ahel et al, 2009; Chou et al, 2010; Mehrotra et al, 2011; Krietsch et al, 2013; Luijsterburg et al, 2016; Teloni & Altmeyer, 2016; Gupte et al, 2017).

ADPr at DNA damage sites is then efficiently removed to facilitate the timely repair by allowing access to downstream repair factors and preventing trapping of PARP1 and DNA repair factors at the sites of damage. In addition, removal of PAR prevents excessive PAR accumulation which can lead to free PAR formation, release of apoptosis inducing factor (AIF) from mitochondria, and induction of cell death via the parthanatos pathway (Wang et al, 2009; Abplanalp & Hottiger, 2017). Two (ADP-ribosyl)hydrolase enzymes are responsible for the reversal of most of the DNA damage induced by ADPr, PARG, and ARH3. PARG is highly efficient at hydrolyzing long PAR chains but cannot remove serine-linked MAR (Lin et al, 1997; Slade et al, 2011; Rack et al, 2021). ARH3 is the only known enzyme with an activity against serine-linked MAR synthesized by the PARP1/2–HPF1 complex under both stressed and unstressed

[1]Translational Neurosciences, Faculty of Medicine and Health Sciences, University of Antwerp, Antwerp, Belgium  [2]Laboratory of Neuromuscular Pathology, Institute Born-Bunge, University of Antwerp, Antwerp, Belgium  [3]Sir William Dunn School of Pathology, Oxford University, Oxford, UK  [4]Department of Pediatric Neurology, Antwerp University Hospital, Antwerp, Belgium  [5]Neurology Clinic, Clinical Center of Serbia, Faculty of Medicine, University of Belgrade, Belgrade, Serbia  [6]Clinic for Neurology and Psychiatry for Children and Youth, Faculty of Medicine, University of Belgrade, Belgrade, Serbia  [7]Neuromuscular Reference Centre, Department of Neurology, Antwerp University Hospital, Antwerp, Belgium

Correspondence: Jonathan.Baets@uantwerpen.be; ivan.ahel@path.ox.ac.uk
Danique Beijer and Thomas Agnew are shared first author
Ivan Ahel and Jonathan Baets are shared last author

conditions (Fontana et al, 2017; Palazzo et al, 2018). ARH3 was shown to possess PAR degrading activity albeit much lower than PARG, hence PARG is the main enzyme controlling PAR dynamics in human cells (Oka et al, 2006; Fontana et al, 2017; Rack et al, 2021).

Many of the enzymes involved in the regulation of ADPr, an important pathway controlling cellular stress, are ubiquitously expressed. Despite this, dysregulation of cellular ADPr has been primarily implicated in the pathogenesis of neurological disorders most commonly neurodegeneration (Hanai et al, 2004; Chiarugi, 2005; Hu et al, 2017; Ghosh et al, 2018; Wang et al, 2018a). Elevated PARP and PAR has been reported in the frontal and temporal lobes of the brains of Alzheimer patients (Love et al, 1999). Similarly, increased PARylation has been noted in several other neurodegenerative disorders, including Parkinson's disease, amyotrophic lateral sclerosis, and Huntington's disease (Kim et al, 2004; Vis et al, 2005; Kam et al, 2018; McGurk et al, 2018). Whereas no pathogenic mutations in either PARG or PARP1 have been reported in connection with neurodegeneration to date, mutations in several other key regulators of cellular ADPr have been shown to be causal for a spectrum of neurodegenerative phenotypes. Recessive mutations in TARG1 cause a severe neurodegenerative disorder with seizures without dysmorphic features (Sharifi et al, 2013). X-linked hemizygous mutations in AIFM1, which encodes apoptosis-inducible factor (AIF) a binder of free PAR, are causative for Charcot–Marie–Tooth disease type 4 in combination with deafness and cognitive impairment (Hu et al, 2017; Wang et al, 2018a).

Recently, recessive ADPRHL2 (ADPRS) mutations, encoding the ARH3 protein, were shown to cause a congenital-onset neurodegenerative stress-induced (epileptic) ataxia syndrome with early pediatric onset (CONDSIAS) (Danhauser et al, 2018; Ghosh et al, 2018; Aryan et al, 2020). The neuronal vulnerability in humans is mirrored in model organisms where knockouts of a Parg homolog in Drosophila melanogaster leads to larval-stage lethality but when grown at permissive temperatures show reduced locomotion, PAR accumulation, global neurodegeneration, and premature death (Hanai et al, 2004). Despite the observed association between dysregulation of cellular ADPr or loss-of-function mutations in ARH3 and neurodegeneration, the mechanistic underpinning of the observed tissue-specific vulnerability of neurons is not yet fully clear. ARH3-deficient MEFs and, recently, patient-derived fibroblasts have been shown to have increased sensitivity to hydrogen peroxide ($H_2O_2$) which can be rescued by PARP inhibition, suggesting that ARH3 activity is especially required during periods of cellular stress (Mashimo et al, 2019; Prokhorova et al, 2021a). The function of ARH3 in the regulation of ADPr has been most extensively studied in the context of nuclear DNA damage repair pathways and the cellular responses to PARP inhibitors (Fontana et al, 2017; Wang et al, 2018b; Prokhorova et al, 2021a, 2021b). However, it is unclear whether deficiencies in this pathway are the primary driver of neurodegeneration in patients with ARH3 loss-of-function mutations or whether loss of cytosolic or mitochondrial ARH3 activity contributes to disease progression (Mashimo et al, 2013; Fontana et al, 2017; Prokhorova et al, 2021a). Furthermore, ADPr also has an important role in the regulation of chromatin structure and histone modifications as histone H3 Ser10 ADPr has been shown to be mutually exclusive with neighboring Lys9 acetylation (Bartlett et al, 2018; Prokhorova et al, 2021a). Interestingly, deficiencies in ARH3 activity in patient cells indeed results in persistent MAR on histones altering canonical histone marks, most notably H3K9Ac,

thereby causing altered transcription and subsequent cellular dysfunction (Bartlett et al, 2018; Hanzlikova et al, 2020; Prokhorova et al, 2021a).

In this study, we present two families with (complex) juvenile-onset neurodegenerative peripheral neuropathy with homozygous ADPRHL2 mutations. The identification of these families expands the clinical phenotype associated with ADPRHL2 mutations from a neurodegenerative central nervous system phenotype with ataxia, epilepsy and neurodevelopmental delay (CONDSIAS) to a phenotype that has a striking peripheral neuropathy as its core component among several other features. Furthermore, characterization of the novel C26F variant and the recurrent V335G variant demonstrate a division in the underlying mechanisms of ADPRHL2 mutations in which the C26F mutation results in a complete loss of functional protein. Conversely, the V335G mutation results in mild loss of expression, similar to other ADPRHL2 mutations reported previously, while retaining enzymatic activity with a distinct loss of nuclear localization. Despite the reduced nuclear localization of the V335G mutant protein, the basal ADPr levels remain unchanged and nuclear ADPr is highly elevated after stress implicating the loss of nuclear ARH3 activity following cellular stress in the pathology of patients with ADPRHL2 mutations resulting in CONDSIAS or related disorders.

# Results

## Clinical description of patients

The patients in this study were included in a larger hereditary motor neuropathy cohort of 73 families for whom next-generation sequencing (NGS) was performed to identify the genetic origin of their disorder.

Both patients in family A were born at term after uneventful pregnancies and the parents reported no known consanguinity. For patient A:II:1, early psychomotor and general development was normal. At age 13 he presented with walking instability. The overall progression was slow. At age 33, he has walking instability and is unable to walk on the heels. There is moderate atrophy of the distal third of his legs and arms. He reports mild sensory involvement in the form of hypoesthesia in tip toes, deep position, and a severely diminished vibration sensitivity in his legs. Additional clinical features include micrognathia, nystagmus, postural tremor, and mild leg spasticity, indicating upper motor neuron involvement. The brain MRI performed at age 13 yr was normal. The EMG results are consistent with mild axonal predominantly motor polyneuropathy. Brain MRI at the age of 33 yr showed mild white matter hyperintensities.

Like his brother, early psychomotor and general development for patient A:II:2 was normal. He presented at 15 yr of age with fatigue and instability during walking. The overall progression is slow. At age 31 yr, he had mild gait ataxia and instability and showed moderate atrophy of distal parts of upper and lower limbs as well as mild atrophy in the proximal regions. Sensory involvement consists of hypoesthesia and reduced vibration sense in his legs. Like his brother he shows a nystagmus and mild leg spasticity. In addition, he shows a postural tremor and myoclonic jerks. Brain MRI performed at age 26 yr was normal. The EMG results are consistent with mild axonal predominantly motor polyneuropathy. This patient passed

away unexpectedly due to an infectious syndrome followed by rapid onset respiratory failure and cardiac arrest.

The proband in family B, was born to consanguineous parents after an uneventful pregnancy and at term delivery. At 15 mo of age, she presented with febrile seizures. She developed normal but somewhat delayed speech and moderate intellectual disability by the age of 6 yr. She showed growth retardation, moderate scoliosis, and *pes cavus*. She suffered from exostosis, which are explained by the known *EXT1* variant NM_000127.2 (EXT1): c.538_539delAG (p.Leu181Profs*7). The *EXT1* variant is also present in her father whose sole symptoms consist of an exostosis phenotype. From the age of 10 yr, she developed moderate atrophy of intrinsic hand muscles and mild atrophy of the distal lower limbs. Weakness of foot dorsiflexors resulted in a drop foot. No sensory involvement was reported. Weakness was clearly more pronounced in the upper limbs than in the lower limbs. Brain MRI performed at age 13 was normal. The EEG showed sporadic epileptiform activity in frontocentral localization, without clinical seizures. The EMG showed a pronounced axonal motor polyneuropathy, with normal sensory parameters. Treatment with gammaglobulins temporarily ameliorated her motor regression. The patient passed away because of respiratory failure at the age of 16 yr. The clinical findings for all three patients are summarized in Table 1.

### Identification of a novel and a recurrent *ADPRHL2* mutation

NGS data acquired for two patients and the parents in family A and the proband and father in family B were analyzed. No variants in genes known for neuropathy or associated disorders were withheld for potential causality after segregation analysis. Subsequent analysis of genes not yet associated with neuropathies identified homozygous missense mutations in *ADPRHL2* in all three patients.

In family A we identified NM_017825: c.1004T>G (p.Val335Gly), this variant has since been reported by Danhauser et al (2018). In family B, we identified NM_017825: c.77G>T (p.Cys26Phe). Validation of mutations was performed using di-deoxy sequencing. Both mutations were homozygous in the patients (Fig 1). For family A, the unaffected parents are confirmed heterozygous. In family B, the father is confirmed heterozygous, but we were unable to obtain DNA from the mother.

### Variable loss of protein stability for ARH3 mutants

To investigate the effects of the identified mutations (NM_017825: c.77G>T: p.Cys26Phe and NM_017825: c.1004T>G: p.Val335Gly, referred to as C26F and V335G, respectively) on ARH3 protein structure and function we first expressed and purified the ARH3 variant proteins in *Escherichia coli*. Mutant and WT expression vectors were created as previously described (Fontana et al, 2017). We also purified and tested a previously characterized catalytic null variant of ARH3 (D77N D78N) to act as a negative control. All proteins showed robust overexpression in whole cell lysates (wcls). However, recovery after lysis varied: WT and D77N D78N were clearly present in the soluble fractions, V335G showed somewhat reduced stability while C26F was completely insoluble suggesting the C26F amino acid substitution causes misfolding or aggregation of the protein (Fig 2A).

Structural analysis of the mutant positions shows that Cys26 is located in the protein core facing the center of a four α-helical bundle (Fig 2B). The increase in Van der Waals volume associated with the C26F mutant will likely disrupt this packing, induce misfolding, and thus causes the observed reduction in solubility and stability. Val335 is positioned within a surface loop and faces a hydrophobic pocket, thus contributing to anchoring the loop to the main protein body (Fig 2B). The V335G mutation removes this interaction, probably causing exposure of the hydrophobic pocket, which may cause the mild reduction in overall structural stability.

### V335G mutant retains enzymatic activity in vitro

To investigate the enzymatic activity of the purified ARH3 protein variants, we performed an end point in vitro serine-(ADP-ribosyl) hydrolase activity assay using radiolabelled MARylated histone H3 peptide (H3$^{MAR}$) as a substrate, as previously described by Fontana et al (2017). H3$^{MAR}$ was synthesized by incubating histone H3 (H3) peptide with PARP1–HPF1 complex in the presence of $^{32}$P-NAD$^+$. The reaction was stopped using the PARP inhibitor olaparib. Radiolabeled H3 peptide was then incubated with ARH3(WT), ARH3(D77N D78N), or ARH3(V335G) protein as described in the Materials and Methods section. Consistent with previous results, ARH3(WT) protein removed radiolabelled ADP-ribose from H3$^{MAR}$ peptide and conversely the catalytic dead mutant of ARH3(D77N D78N) showed no observable hydrolase activity against the H3$^{MAR}$ peptide consistent with published results (Fig 2C). Interestingly, ARH3(V335G) protein showed comparable activity against H3$^{MAR}$ as ARH3(WT) protein showing that the V335G amino acid substitution does not negatively influence the enzymatic activity of ARH3 in vitro in this end point assay. Further kinetic analysis is required to better determine whether the enzymatic activity of the mutant ARH3(V335G) differ to that of ARH3(WT) protein. These results indicate that that there may be more than one mechanism by which mutations in *ADPRHL2* are pathogenic. Because of the insolubility of the ARH3(C26F) protein we were unable to assess its enzymatic function in this assay. In contrast, the V335G mutant does retain significant solubility and retains enzymatic function indicating that the pathogenicity of the V335G amino acid substitution may be independent of enzymatic activity in vivo and occurs via an alternate mechanism.

### Variable loss of ARH3 protein expression in patient fibroblasts

We were able to acquire fibroblast cell lines from two of the *ADPRHL2*-patients and an unrelated healthy control individual. Using these cell lines, we aimed to assess wild-type and mutant ARH3 expression and subcellular localization in patient cells. Consistent with our in vitro data, no ARH3 protein was observed in C26F patients cells, and while it was possible to detect some ARH3 protein in V335G patient cells, steady state protein levels were substantially reduced when compared with controls (Fig 3A). To determine whether the reduction in mutant ARH3 protein was due to reduced protein solubility, as opposed to reduced expression, we performed cellular fractionation of soluble and insoluble fractions. From these experiments we observed that the lack of protein expression for the V335G and C26F mutant was likely not due to a reduction in the solubility of ARH3 as all protein in V335G cells was in the soluble fraction (Fig 3B). Taken together with the the in vitro data,

**Table 1. Clinical description of patients carrying homozygous *ADPRHL2* missense variants showing variable phenotypes.**

| Individual | A:II:1 (patient 1) | A:II:2 (patient 2) | B:II:1 (patient 3) |
|---|---|---|---|
| Gender | M | M | F |
| Parental consanguinity | Reported negative | Reported negative | + |
| Current age or age at death | 34 yr | 32 yr[a] | 16 yr[a] |
| Circumstances of death | — | Cardiac arrest/respiratory failure | Respiratory failure |
| *ADPRHL2* Mutation | | | |
| Genomic position (hg19) | Chr1: 36558899T>G | Chr1: 36558899T>G | Chr1: 36554582G>T |
| cDNA | NM_017825: c.1004T>G | NM_017825: c.1004T>G | NM_017825: c.77G>T |
| Protein | p.Val335Gly | p.Val335Gly | p.Cys26Phe |
| Clinical features | | | |
| Age at onset | 13 yr | 15 yr | 15 mo |
| Symptoms at onset | Walking instability and intermittent lateropulsion | Fatigue and instability during walking | Febrile seizures |
| Psychomotor development | Normal | Normal | Normal speech, moderate intellectual disability (6 yr) |
| General development | Normal | Normal | Growth retardation for which growth hormones were supplied |
| Gait | Weakness of foot dorsiflexors, drop foot, and mild spasticity | Foot dorsiflexor weakness, drop foot, mild spasticity, and instability; later also affected by fracture | Weakness of foot dorsiflexors, drop foot |
| Muscle atrophy | Moderate atrophy of distal third of upper and lower limbs | Moderate atrophy of distal upper and lower limbs and mild proximal atrophy | Moderate atrophy of intrinsic hand muscles (10 yr), mild atrophy of distal lower limbs |
| Proximal strength upper limb | 5 | 5 | 5 |
| Distal strength upper limb | 4 | 2–4 | 2/5 to 4-/5 |
| Proximal strength lower limb | 5 | 5 | 5 |
| Distal strength lower limb | 1–2 | 1–3 | 4-/5 to 5/5 |
| Reflexes upper limb | Normal | Diminished | Normal |
| Reflexes lower limb | Normal | Distally diminished | Normal |
| Sensory involvement | Hypoesthesia in tip toes, deep position, and vibration sense severely diminished in lower legs and hands | Hypoesthesia and loss of vibration sense in legs | - |
| Seizure type | - | Myoclonic jerks | Febrile seizures |
| Cardiac features | Normal | Normal | Left ventricle hypertrophy and mitral insufficiency |
| Other clinical features | Motor tics in childhood, micrognathia, nystagmus, postural tremor, absent trunk hair, pes cavus, mild to moderately restrictive pulmonary function, and scoliosis | Nystagmus, postural tremor, mild dysarthria, pes cavus, hyperhidrosis, absent trunk hair, carpal tunnel surgery, and mixed restrictive/obstructive lung function | Moderate scoliosis, growth retardation, pes cavus, and exostosis with confirmed causal EXT1 variant |
| Neurological examination | | | |
| EMG | Severe axonal motor polyneuropathy and mild sensory involvement | Severe axonal motor polyneuropathy and mild sensory involvement | Profound axonal motor polyneuropathy, no sensory involvement |

**Table 1. Continued**

| Individual | A:II:1 (patient 1) | A:II:2 (patient 2) | B:II:1 (patient 3) |
|---|---|---|---|
| Brain MRI (age performed) | Normal (13 yr) | Normal (26 yr) | Normal (13 yr) |
|  | Mild white matter hyperintensity lesions (33 yr) |  |  |
| EEG | Normal | Mild nonspecific changes with intermittent bifrontal theta waves | Sporadic epileptiform activity frontocentral localization |
| Other genetic features | – | – | NM_000127.2 (EXT1): c.538_539delAG (p.Leu181Profs) |

[a]Individual is deceased

these results show that the C26F amino acid substitution is destabilizing and results in an undetectable level of ARH3 protein and therefore cellular ARH3 enzymatic activity. The V335G allele, results in a reduction in the steady state levels of ARH3 protein but the protein that is present is soluble and is likely enzymatically active based on our in vitro assay. The overall ARH3 activity in V335G cells is therefore likely reduced but not absent when compared with control cells with WT ARH3 function.

## Loss of mutant V335G ARH3 protein in nuclear and cytoplasmic compartments

ARH3 protein has been described to function in several different pathways in human cells and localized to several different sub-cellular compartments. ARH3 protein has been shown to reside in the cytosol, the nucleus and in mitochondria. It is currently not yet known how the sub-cellular distribution of ARH3 protein is regulated, although ARH3 has been shown to rapidly recruit to DNA lesions generated by laser micro-irradiations in a PARP1 dependent manner (Wang et al, 2018b). The neurodegenerative phenotypes associated with loss-of-function mutations in ARH3 may therefore be due to loss of ARH3 activity in nuclear, mitochondrial, or cytosolic

pathways or, indeed, a combined loss of ARH3 activity throughout the cell.

Our results showed that ARH3(V335G) mutant protein is catalytically active but with reduced steady state protein levels. To better understand the pathology associated with this interesting hypomorphic allele, we aimed to characterize the sub-cellular localization and function of ARH3(V335G) mutation protein in patient cells.

We tested several commercially available ARH3 antibodies to see whether they specifically recognize ARH3 protein via indirect immunofluorescence; however, none of them were suitable (data not shown). We did, however, identify a specific ARH3 antibody suitable for Western blotting using U2OS ARH3 KO cells as a negative control (Fig S1). We performed cellular fractionation and subsequent immunoblot to determine the subcellular distribution of ARH3 protein in ARH3(V335G) patient cells compared with control cells (Fig 3C–E). Immunoblotting the different subcellular protein fractions for the WT and V335G mutant showed a marked difference in ARH3 localization in the cytosolic fraction, with relatively unchanged expression in mitochondria and nuclei (Fig 3C and D). This result indicates that the loss of steady state levels of ARH3 protein in wcl from ARH3(V335G) patient cells compared to controls is due to dysregulated turnover of cytosolic ARH3 protein rather than a reduction in the stability of ARH3 protein overall otherwise one would expect ARH3 protein to be reduced by equal amounts in all cellular fractions. Furthermore, this result suggests the pathology observed in patients with the V335G allele is due to loss of cytosolic ARH3 function and is not due to loss of mitochondrial ARH3 function (Fig 3C and D).

We performed live cell imaging using ARH3(WT)-GFP, ARH3(D77N D78N)-GFP, and ARH3(V335G)-GFP expressed in human U2OS cells, in which we assessed co-localization with the nuclear and mitochondrial compartments using Hoechst and MitoID, respectively (Fig 4). Both ARH3(WT)-GFP and ARH3(D77N D78N)-GFP localize to the nucleus, cytosol, and mitochondria. Strikingly, ARH3(V335G)-GFP showed a substantial reduction in the nuclear signal compared with ARH3(WT)-GFP (Fig 4). Repeated experiments using BE(2)-M17 neuroblastoma cell lines showed similar results (Fig S2A). Whereas this result in overexpression models using osteosarcoma U2OS and neuroblastoma BE(2)-M17 cells somewhat differs from the cellular fractionation results in patient (fibroblast) cell lines, where nuclear ARH3 protein was unchanged but cytosolic ARH3 was lost, it demonstrates that dysregulation of the subcellular localization of ARH3(V335G)-GFP is due to the mutation rather than loss of catalytic

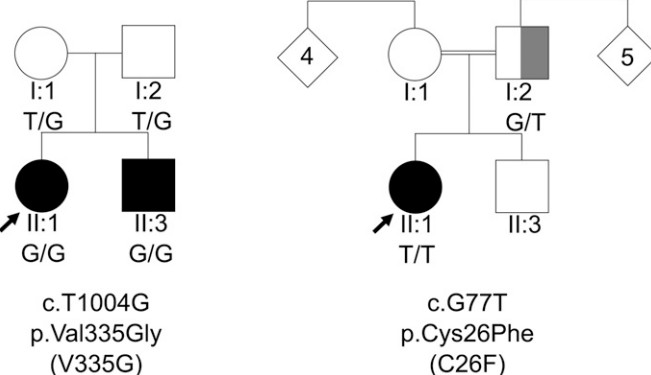

**Figure 1. Autosomal recessive inheritance of ADPRHL2 mutations in two hereditary motor neuropathy families.**
Pedigrees of families A and B with their respective mutation and the segregation of each by genotype, showing affected (black), unaffected (white). The patient and the partially affected parent (grey) in family B, both carry a known causal *EXT1* variant causal for their exostosis phenotype. The father does not present with the hereditary motor neuropathy and neurodevelopmental phenotype.

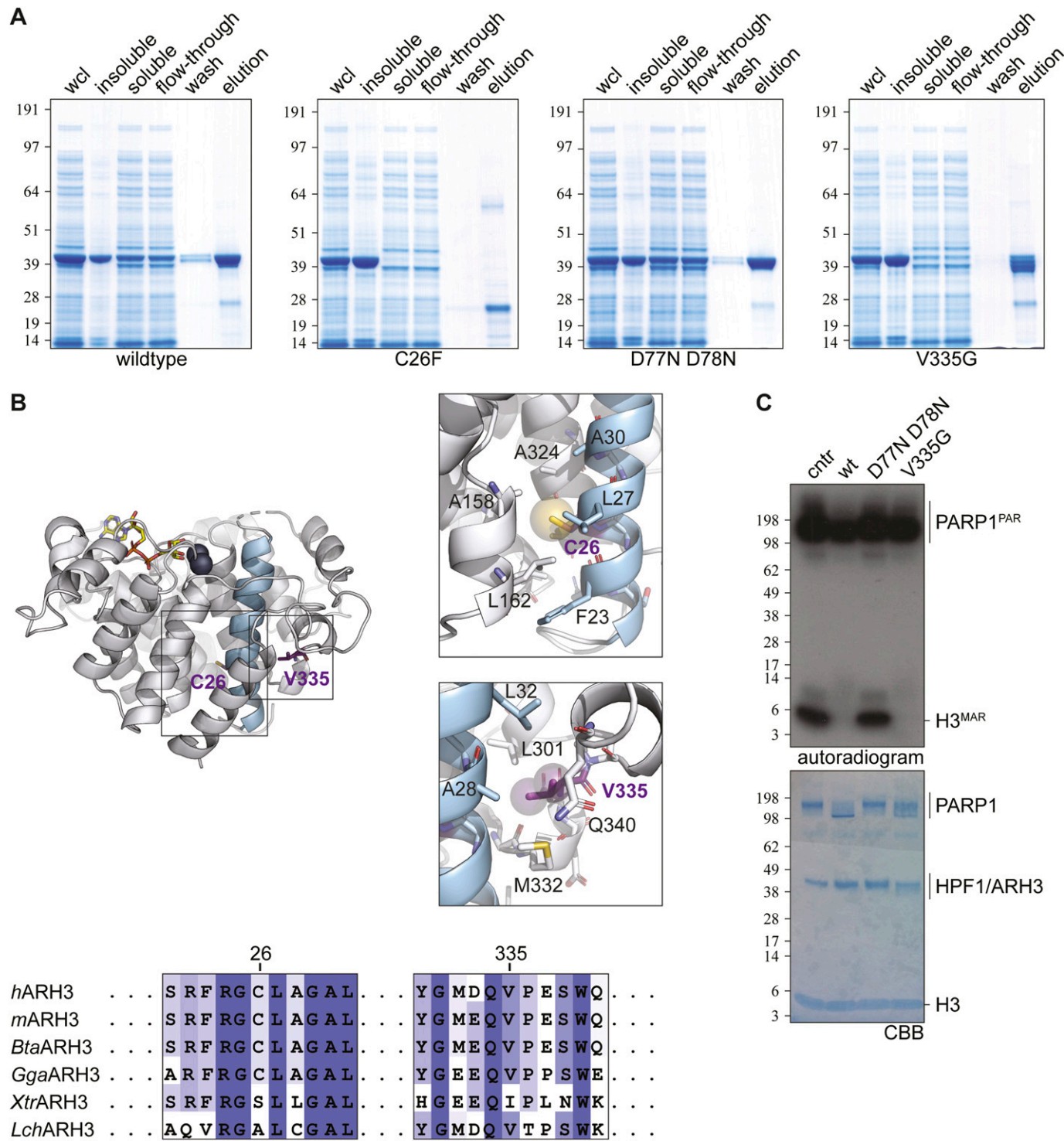

**Figure 2. In vitro expression and activity of ARH3 (mutant) protein and ribbon representation of ARH3 in complex with ADP ribosylation (yellow) and Mg²⁺ ions (dark blue).**

**(A)** SDS–PAGE analysis of expression and purification of recombinant ARH3 wild type and mutants in *Escherichia coli*. ARH3 (theoretical Mw 42.88 kD) was enrich from whole cell lysate by nickel affinity chromatography (for details, see the Materials and Methods section). Both C26F and V335G show similar expression, but lower abundance in the soluble fraction, compared with WT and D77N D778N mutant. **(B)** Alpha-helix 1, containing Cys26, is highlighted for orientation purposes. Right panels: Van der Waals radii of Cys26 sulphur and Val335 side chain carbon atoms are depicted as transparent spheres. Residue Cys26 is located in the core of a conserved helical bundle (right upper panel). Positioning of this residue within the structure suggest that the increase in Van der Waals volume associated with the C26F mutation incompatible with correct packing. Residue Val335 is located in partial structured surface loop packing against α-helix 1 (right lower panel) and is inserted in a hydrophobic pocket. The structural consequences of the V335G mutation are not immediately appreciable but may weaken the local packing, expose hydrophobic

activity (Fig 3). Specifically, transport of ARH3 between nucleus and cytosol appears to be affected, whereas mitochondrial localization is not. The unchanged levels of cytosolic ARH3(V335G)-GFP observed using our over-expression system is likely explained by the rate of protein synthesis being greater than the rate of degradation as it is controlled of a strong promotor which is likely not the case for the endogenous protein.

**V335G patient cells accumulate nuclear ADPr after DNA damage**

The best characterized function of ARH3 in human cells, to date, is within the nuclear DNA damage response, as ARH3 has been shown to be recruited to DNA lesions in a PARP1-dependent manner and fails to recruit following pre-treatment with the PARP1/2 inhibitor olaparib (Wang et al, 2018b). Lack of ARH3 was recently shown to leave post-repair ADP-ribosyl "scars" on the chromatin, which can influence nearby transcription (Bartlett et al, 2018; Hanzlikova et al, 2020; Prokhorova et al, 2021a). We tested the functions of ARH3(V335G) in the nuclear DNA damage response in human cells to determine whether this function of ARH3 is retained in patients with V335G mutation. As previously shown, there are reduced levels of nuclear ARH3(V335G)-GFP protein; however, we wanted to examine whether cytosolic ARH3(V335G)-GFP protein could translocate into the nucleus after DNA damage as has been shown for several DNA-damage repair proteins such as APLF (Mehrotra et al, 2011). To this end, we transiently transfected human U2OS cells with ARH3(WT)-GFP or ARH3(V335G)-GFP subjected to laser micro-irradiation coupled to live-cell imaging (Fig S2B and C). Consistent with previously published data, ARH3(WT)-GFP was efficiently recruited to laser-stripe induced DNA damage sites in the nucleus (Wang et al, 2018b). ARH3(V335G)-GFP was not recruited to laser stripe induced DNA damage lesions indicating that the nuclear role of ARH3 is dysfunctional in patients homozygous for the ARH3 V335G allele (Fig S2B and C).

To test ARH3 activity within the DNA damage response in patient cells, where ARH3 protein levels are endogenously controlled, we treated WT and V335G cells with the DNA damaging agent $H_2O_2$ and repeated cellular fractionation. After $H_2O_2$, a small reduction in total ARH3 protein was observed in V335G patient cells; however, this was also observed in control cells (Fig S3). Most importantly, these results clearly show that after cellular stress, nuclear ADPr is increased in patient cells compared with control cells but remained unchanged in cytosolic and mitochondrial fractions (Fig 5A and C). Furthermore, there is an increase in nuclear ARH3 protein in wild-type cells compared with V335G patient cells, indicating that the observed increase in $H_2O_2$ induced nuclear ADPr in patient cells is due to reduced levels of nuclear ARH3 (Fig 5A–C). It is likely that the reduced levels of nuclear ARH3 in patient cells is, at least in part, due to a reduction in the pool of cytosolic ARH3 protein. Alternatively, it could be that ARH3 shuttling is impaired specifically by the V335G mutation as this residue is located on the surface of the

ARH3 protein (Fig S2B and C) Overall, these results show that despite enzymatically active ARH3 present in V335G patient cells, the recruitment of ARH3 to DNA damage lesions is dysfunctional due both depleted ARH3 steady state levels and furthermore dysregulated nuclear import of ARH3 following genotoxic stresses. Conversely, evidence from cellular fractionation experiments in patient cells and from over-expression experiments indicate that mitochondrial ARH3 localization and steady state levels are unaffected by the V335G amino acid substitution indicating that the neurological pathology associated with the V335G allele is not due to loss of ARH3 function in mitochondria.

# Discussion

In this study, we present two families with homozygous mutations in *ADPRHL2* (encoding ARH3 protein) with a juvenile-onset complex phenotype dominated by a peripheral neuropathy. The missense mutations we identified have reduced protein stability in the case of the C26F mutant or altered subcellular localization in case of the V335G mutant.

Recessive mutations in *ADPRHL2* were previously identified in patients with a childhood degenerative epileptic ataxia syndrome and patients with developmental delay, ataxia, and axonal neuropathy (Danhauser et al, 2018; Ghosh et al, 2018). The patients described by Ghosh et al (2018), are asymptomatic early after birth, but gradually develop infection-related spontaneous epileptic seizures or present with a neurodegenerative course including weakness, ataxia, loss of milestones, and further clinical deterioration that ultimately leads to premature death. In contrast, Danhauser et al (2018), reported a spectrum of patients with different combinations of symptoms including developmental delay, ataxia, seizures, sensorimotor neuropathy, hearing loss, respiratory insufficiency, and structural brain defects (Danhauser et al, 2018).

The phenotype we observed in patients with *ADPRHL2* mutations broadens the spectrum into peripheral nervous system (PNS) predominant disorders. Our patients were included in the NGS study based on the presence of a profound peripheral motor neuropathy, and for two of three patients, this remained the most prominent feature after evolution of the disease over many years. In addition to the distal axonal neuropathy, patient 1 and 2 both show mild postural tremor, nystagmus, and leg spasticity indicating mild CNS involvement. Patient 3 presents with a clinical phenotype that has some similarities with the pediatric epileptic ataxia syndrome (CONDSIAS) previously reported, as her initial symptoms include febrile seizures and intellectual disability. Despite not presenting with epileptic seizures later on in life, her EEG also showed sporadic epileptiform activity. She has moderate scoliosis and like the other two patients also a profound distal

---

residues and thus affect the overall structural stability of the protein. Note that in the right panels foreground structural elements have been removed to allow representation of the buried residue pockets. Image was created with PyMOL v2.3 (Schrodinger LLC) using human ARH3 in complex with ADP-ribose (PDB 6D36). **(C)** The (ADP-ribosyl)hydrolase activity of ARH3 WT and mutants was assessed using H3 and poly(ADP-ribose)polymerase (PARP)1 MARylated and PARylated, respectively, in presence of $^{32}$P-NAD$^+$ as substrates. After the reaction samples were analyzed by autoradiogram and SDS–PAGE. Both WT and V335G were active under the assay conditions. cntr (control; no ARH3).

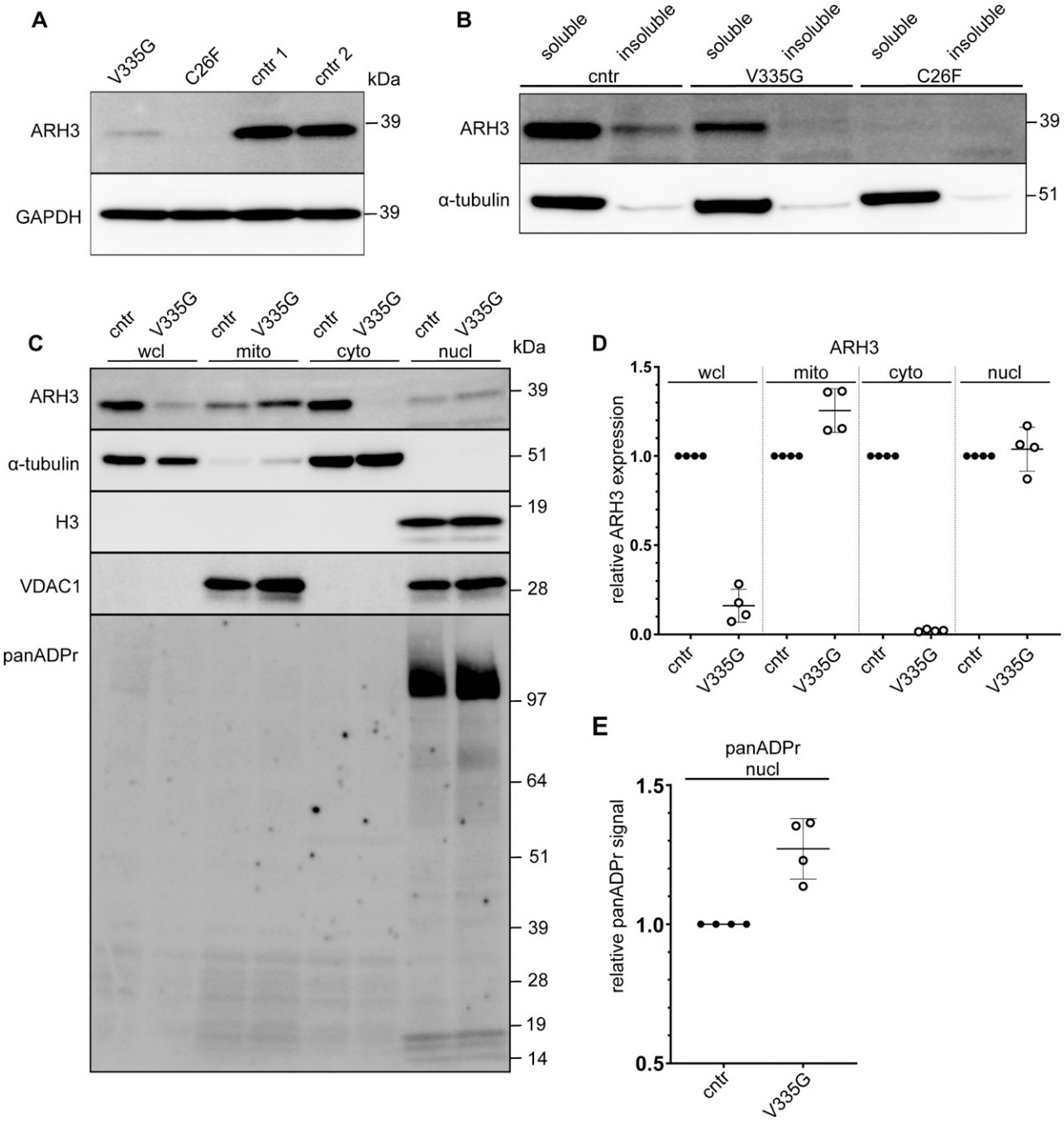

**Figure 3. Protein expression in control, C26F and V335G patient fibroblast cells.**
**(A, B, C)** Whole-cell lysates of patient-derived and control (cntr: healthy individual) fibroblast immunoblotted for ARH3 and GAPH (loading control) for the different mutations V335G (family A), C26F (family B) (B) soluble and insoluble whole-cell lysate fractions of patient and control fibroblasts immunoblotted for ARH3 and α-tubulin (loading control) (C) subcellular fractions of patient and control fibroblasts immunoblotted for α-tubulin (cytosolic control), VDAC1 (mitochondrial control) and Histone H3 (H3) (nuclear control) in whole cell lysate, mitochondrial fraction (mito), cytosolic fraction (cyto), and nuclear fraction (nucl). **(D, E)** Quantification of ARH3 expression of V335G mutant relative to control (cntr: healthy individual) per fraction normalized to the respective subcellular fraction control α-tubulin/H3/α-tubulin/VDAC1 showing (n = 4, mean and SD) (E) quantification of panADPr signal of V335G mutant relative to control (cntr: healthy individual) per fraction normalized to the respective subcellular fraction control α-tubulin/H3 (n = 4, mean and SD).

motor axonal neuropathy. Of note is the fact that for two patients in the current study rapid clinical deterioration with respiratory insufficiency and premature death was observed, possibly triggered by an intercurrent infection and fever. Such rapid deterioration is rather unusual in other inherited peripheral neuropathies. This could be relevant to the normal function of *ADPRHL2* in stress-response although more clinical observations are likely needed to confirm this.

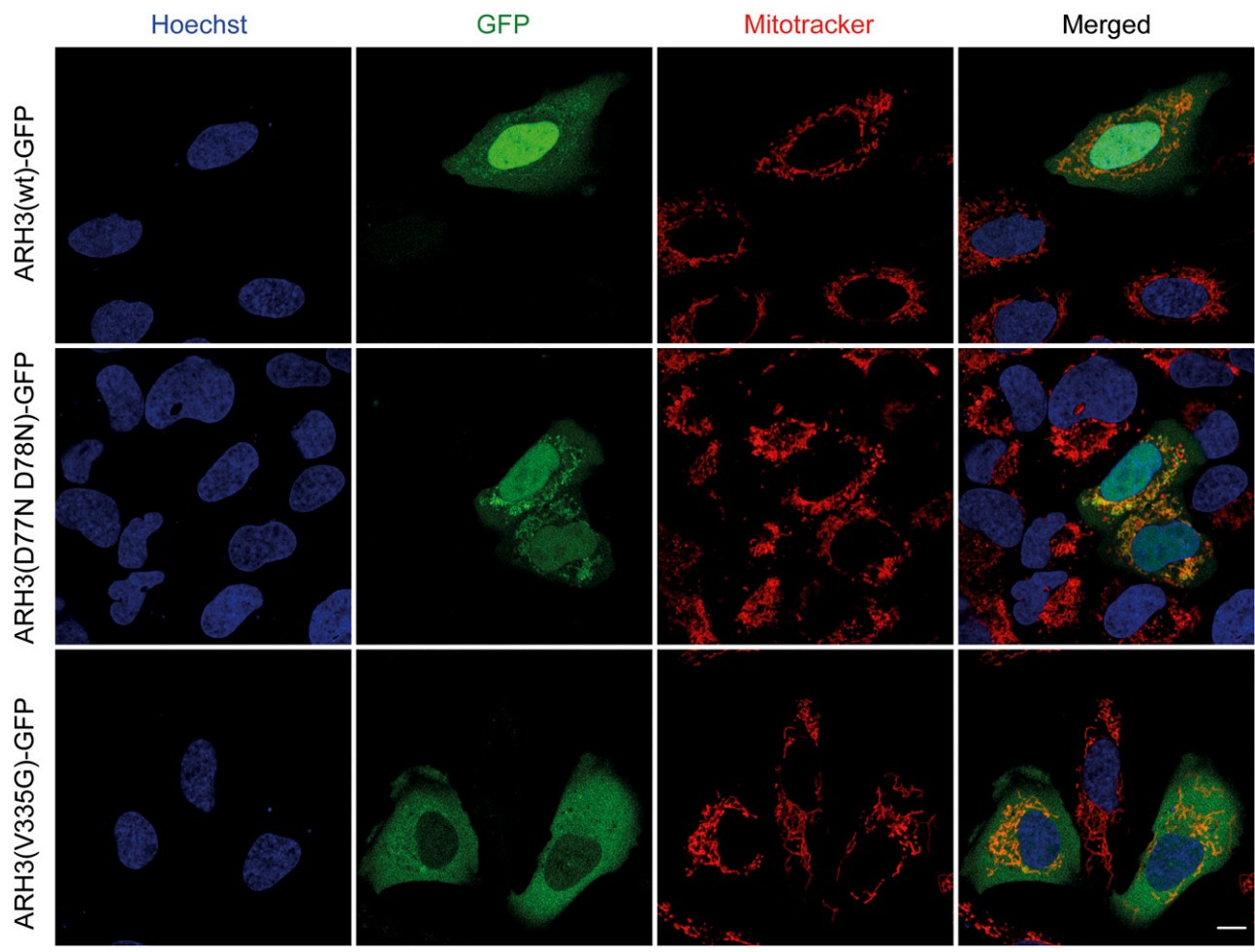

**Figure 4. Live cell imaging of U2OS overexpression model.**
Localization of ARH3 protein as checked by live cell imaging for GFP-tagged ARH3 (green) with mitotracker (red) and Hoechst (blue) staining in separate and merged images in ARH3(WT)-GFP wild-type and mutants D77N D78N (catalytic null) and V335G in transfected U2OS cells. Size bar indicates 10 μm.

Variability in phenotypes were already observed by Ghosh et al (2018), and Danhauser et al (2018), where, for example, the seizure phenotype varies from generalized tonic-clonic seizure with an onset at 9 mo, to patients without seizures at all (Danhauser et al, 2018; Ghosh et al, 2018). Furthermore, cerebellar atrophy, intellectual disability, ataxia and motor axonal neuropathy was observed in several but not all patients. Many of the reported patients, as well as our patient 3, come from consanguineous families. Consanguinity in the parents increases the chances of multiple recessive diseases in the same patient, a so-called *double-trouble* or *double hit* phenotype. This could potentially account for some of the diversity of the observed phenotypes (Hodapp et al, 2006; Goldenberg-Cohen et al, 2013; Gonzaga-Jauregui et al, 2015; Bis-Brewer et al, 2020). The recurrent V335G mutation seems causative for a variable phenotype as observed in several families suggesting that additional modifiers and the other elements of the genetic background of individuals could also account for a proportion of the phenotypic variability observed in *ADPRHL2*-associated disorders (Gonzaga-Jauregui et al, 2015; Bis-Brewer et al, 2020). Although, the effect of such modifiers and more generally genetic background on the phenotypic variability of rare disease phenotypes is difficult to determine, especially for these genetically heterogeneous conditions with a limited number of patients per gene.

In contrast, there is also a possibility that the broad spectrum of *ADPRHL2*-related disease could be explained by the variable underlying mechanisms of the different mutations, or indeed differences in genetic background irrespective of consanguinity or exposure to environmental stressors. We demonstrate this variability in underlying mechanism in this study by investigating the C26F and V335G mutations. We observed that the C26F mutant is highly unstable when over-expressed in *E. coli* and U2OS cells as well as in native patient-derived fibroblasts. In contrast, the V335G mutant shows only a mild stability defect. Similar to the previous report of Danhauser et al (2018), we observe a severe reduction of ARH3 protein for the V335G mutant, however rather than complete loss of protein we still observe detectable amounts of ARH3 in V335G patient fibroblast lines (Danhauser et al, 2018). In addition, we observed similar increased panADPr in V335G fibroblast after $H_2O_2$ exposure as previously shown (Figs 3C and E and 5A and C) (Danhauser et al, 2018). All mutations presented thus far seem to cause at least a severe reduction of ARH3 protein levels; however, it

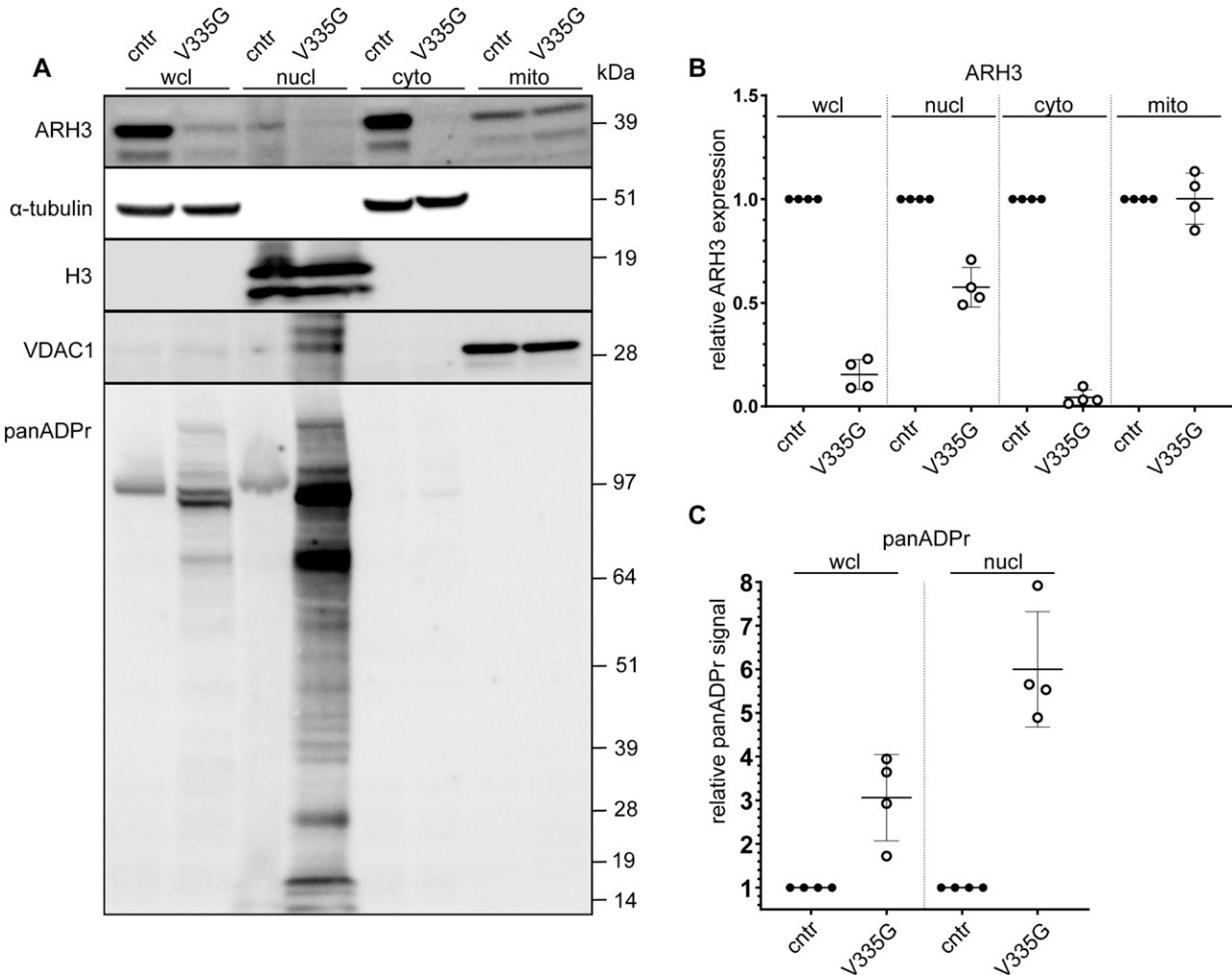

**Figure 5.  Recruitment of ARH3 to DNA damage site.**
**(A)** Subcellular fractions of patient (V335G) and control (cntr: healthy individual) fibroblasts treated with $H_2O_2$ to induce DNA damage, immunoblotted for $\alpha$-tubulin (cytosolic control), H3 (nuclear control) and VDAC1 (mitochondrial control) in whole cell lysate, nuclear fraction (nucl), cytosolic fraction (cyto) and mitochrondial fraction (mito). **(B, C)** Quantification of ARH3 expression of V335G mutant relative to control (cntr: healthy individual) per fraction normalized to the respective subcellular fraction control $\alpha$-tubulin/H3/$\alpha$-tubulin/VDAC1 (n = 4, mean and SD) (C) quantification of panADPr signal of V335G mutant relative to control (cntr: healthy individual) per fraction normalized to the respective subcellular fraction control $\alpha$-tubulin/H3 (n = 4, mean and SD).

is these (potentially) hypomorphic alleles such as V335G and possibly T79P, for which there is some protein detected that might also rely on a different underlying pathomechanism (Danhauser et al, 2018; Ghosh et al, 2018; Aryan et al, 2020).

Variations in (ADP-ribosyl)hydrolase function of ARH3 in the presence of missense mutations have already been shown for mutants created with site-directed mutagenesis targeting known ligand docking regions (Mueller-Dieckmann et al, 2006; Abplanalp et al, 2017; Fontana et al, 2017). Most of these mutants showed loss of hydrolase activity, but the impact of the mutations varied from partial to complete loss of activity. Interestingly, the V335G mutant does not lead to a significantly decreased (ADP-ribosyl)hydrolase activity, hinting towards a different underlying mechanism.

These differences were further validated by intriguing dissimilarity in subcellular localization for the V335G mutant identified in localization assays. Whereas ARH3(WT)-GFP and the catalytic mutant ARH3(D77N D78N)-GFP showed a diffuse distribution through

the nucleus, cytosol, and mitochondria, there is a clear reduction of the ARH3(V335G)-GPF mutant protein in the nucleus and cytoplasm. We were able to corroborate this altered localization of the V335G mutant in patient-derived fibroblasts, which showed a striking reduction in V335G expression in the cytosol but, importantly, mitochondrial ARH3 levels remained unchanged both in patient cell lines and when over-expressed in U2OS cells. Combined with our finding that the ARH3(V355G) retains enzymatic activity, our data strongly suggest that loss of ARH3 activity in mitochondria is not causal of neurodegeneration in patients with loss-of-function ARH3 mutations but is, rather, due to deficiencies in nuclear ARH3 activity. ARH3 has been shown to localize to the nucleus, cytosol and mitochondria but it is not yet fully clear how the subcellular distribution of ARH3 is regulated. The one ARH3 isoform that has been annotated to date has a predicted canonical N-terminal mitochondrial targeting sequence which explains why the mitochondrial targeting of the V335G mutant protein is unaffected.

Mitochondrial dysfunction is a prominent pathomechanism for neurological and in particular neurodegenerative diseases, such as Huntington's disease, Alzheimer's disease, and amyotrophic lateral sclerosis (Stanga et al, 2020). Furthermore, causal genes such as *MFN2* (mitofusin) and the *SLC25* family of mitochondrial carriers, demonstrate a direct link of important mitochondrial processes with neuropathies, neurological and neurodegenerative disease (Züchner et al, 2006; Palmieri et al, 2020). The validation of seemingly unaffected ARH3 mitochondrial function in this mutant is important to differentiate between the mechanism of mitochondrial dysfunction, common in neurodegenerative diseases, and the nuclear DNA damage response effects that are indicated to be causative for *ADPRHL2*-associated diseases (Hanzlikova et al, 2020).

To this date, no nuclear localization sequence has been described for ARH3. The V335G mutation in ARH3 is located in a partially packed surface loop that as we demonstrate does not influence catalytic activity (Figs 2B and 3). It is unclear how the V335G mutation influences the nuclear translocation of ARH3, but it is possible that ARH3 translocates to the nucleus by piggyback mechanisms via interaction with another translocating protein and that the V335G residue located on the surface of ARH3 protein may be required for this interaction. It is the nuclear function of ARH3 that is particularly interesting, as it has been shown that ARH3 is important for hydrolase removal of ADPr from serine residues (Fontana et al, 2017). Moreover, identification of hundreds of DNA damage-induced Ser-ADPr sites in proteins involved in DNA repair, transcription and chromatin organization has revealed that Ser-ADPr sites are the major type of ADPr involved in the regulation of the DNA-damage response (Bonfiglio et al, 2017; Palazzo et al, 2018; Hendriks et al, 2019; Suskiewicz et al, 2020a). In addition, altered or excessive ADPr has an important effect on regulation of chromatin structure and histone modifications, as Ser-ADPr was shown mutually exclusive with acetylation of histone H3 (Bartlett et al, 2018). As such, cellular deficiencies in ARH3, similarly to those in patient cells, result in persistent MAR, also referred to as chromatin scars, on histones which dysregulate neighboring canonical histone modifications, in particular the local histone acetylation (Hanzlikova et al, 2020). These chromatin scars result in altered transcription and subsequent cellular dysfunction (Bartlett et al, 2018; Hanzlikova et al, 2020; Prokhorova et al, 2021a).

We investigated whether the DNA damage response (DDR) is impaired for the ARH3 V335G mutant. We observed that overexpressed ARH3(V335G)-GFP failed to recruit to laser-stripe induced DNA damage sites in U2OS cells and there was increased $H_2O_2$ induced nuclear ADPr in the patient-derived fibroblasts indicating that nuclear DDR is dysregulated in V335G patient cells (Figs 3C and D, 5A and B, and S2B and C). Deficiencies in the DDR have been shown to be associated with neurodegeneration on multiple occasions and there is specific neuronal vulnerability to failing of the ADPr process, with the main focus on PARP1 (Vis et al, 2005; Sharifi et al, 2013; Kam et al, 2018). The mechanism behind this neuronal vulnerability to disturbances in the ubiquitous process of ADPr is still unknown but the post-mitotic nature of neurons implies that damage they acquire might accumulate over time after repeated periods of cellular stress. Similarly, ARH3 acts in conjunction with PARP1, PARG, and several other key regulators of cellular ADPr. The differences in expression levels of each of these players may vary in

neurons compared with other cell types which could account for the increased vulnerability of neurons. Last, it is important to realize that the amount of exposure to stress in terms of DNA damage, infection or different antiviral mechanisms might also influence the extent to which ADPr is required or its dysregulation produces pathology. All these components could partially explain the neuronal vulnerability that is observed with dysregulated ADPr.

The identification of the V335G ARH3 mutant with the profound effect on nuclear localization and DNA damage recruitment demonstrates the importance of the balance and circularity of the ADPr process. The V335G mutant, as opposed to the novel C26F mutant, allows us to study the effects of reduced DNA damage response and possibly in the future to design therapeutic strategies to restore balance to the ADPr process, which previously has only focused on PARP1 inhibition. Striking stress-induced aggravation of the neurological phenotype has been noted as a possible pattern in patients with *ADPRHL2* mutations and this might also be the case even in somewhat less severe PNS predominant phenotypes observed in the current study. It would be interesting to see whether this stress-related aggravation of the symptoms is directly related to the failing of the DDR pathway. In addition, it is important to assess the exact mechanisms by which ADPr in patients with different neurodegenerative disorders results in neurodegeneration, mechanisms that could include $NAD^+$ depletion, accumulation of double-strand DNA breaks, PAR accumulation, and stress granule dynamics.

In conclusion, we have identified two homozygous missense mutations in *ADPRHL2* in two families presenting with a complex motor predominant neuropathy phenotype. In vitro studies of these mutations indicate different underlying pathologies. The C26F mutant protein is unstable, likely leading to a reduced overall activity and expression of ARH3. The V335G mutant retains wild-type (ADP-ribosyl)hydrolase enzymatic activity, has somewhat reduced levels of expression and alters nucleocytosolic subcellular localization without affecting mitochondrial targeting. The relevance of these different pathomechanisms in regard to the patient phenotypes and the overall function of ARH3 within DNA damage response remains to be elucidated further, but our data suggest that loss of nuclear ARH3 function alone is pathogenic.

# Materials and Methods

### Genetic studies

DNA extraction was performed on peripheral blood samples obtained from patients and family members, all patients and/or their legal representatives signed an informed consent, the study was approved by the Ethics Committee of the participating centers. Whole-exome sequencing was performed in family A, whereas whole-genome sequencing was performed for family B. In family A the Nextera Rapid Capture Expanded Exome kit (62 Mb) (Illumina) was used for exome enrichment, in family B the SeqCap EZ Human Exome Library v3.0 (64 Mb) (Roche) was applied. Subsequently, the libraries were sequenced on a HiSeq 2500 platform. For family A, NGS was performed on both probands and parents. For family B, NGS was performed on the proband and father. Annotation and

variant filtering was performed using the Clinical Sequence Analyzer and Miner (Wuxi NextCODE).

No other variants in known neuropathy genes were withheld for potential causality after segregation analysis using di-deoxy sequencing was performed. Subsequently, analysis of genes not yet associated with neuropathies was performed. Co-segregation of variants with the phenotype within each family was confirmed using di-deoxy sequence analysis. Primer sequences are available upon request. Combined annotation dependent depletion was used as a prediction tool for indication of variant pathogenicity.

All genetic and patient data are handled and processed according to the GDPR rules set by the EU, as approved in a Data Management Plan submitted to the University of Antwerp.

### In vitro expression and purification of ARH3

Expression vector for ARH3 was described earlier (Fontana et al, 2017). All indicated mutations were introduced via PCR based site-directed mutagenesis. Rosetta (DE3) cells were grown in LB medium supplemented with 2 mM $MgSO_4$ and antibiotics appropriate for each expression plasmid at 37°C. Expression of recombinant proteins in Rosetta (DE3) cells was induced at $OD_{600}$ 0.6 with 0.4 mM IPTG, cells were grown overnight at 17°C and harvested by centrifugation. Recombinant His-tagged proteins were purified at 4°C by $Ni^{2+}$-NTA chromatography (Jena Bioscience) according to the manufacturer's protocol using the following buffers: all buffers contained 50 mM Tris–HCl (pH 8), 500 mM NaCl, and 10 mM $MgCl_2$; in addition, the lysis buffer contained 25 mM, the washing buffer 40 mM and the elution buffer 500 mM imidazole. All proteins were dialyzed overnight against 50 mM Tris–HCl (pH 8), 200 mM NaCl, 1 mM DTT, and 5% (vol/vol) glycerol. Purity of the protein preparations was assessed using SDS–PAGE and Coomassie brilliant blue staining.

### In vitro (ADP-ribosyl)hydrolase activity assay

ARH3 activity assays were performed essentially as described (Fontana et al, 2017). Briefly, H3 peptide (aa 1–20, biotinylated) was modified by incubation with 0.5 $\mu$M PARP1, 1 $\mu$M HPF1, and activated DNA (Trevigen) in assay buffer (50 mM Tris–HCl [pH 8], 200 mM NaCl, 2 mM $MgCl_2$, 1 mM DTT, 10 $\mu$M $NAD^+$, and 1 $\mu$Ci $^{32}P$-$NAD^+$). Reactions were incubated for 30 min at 30°C and stopped by addition of 1 $\mu$M olaparib. Reactions were further incubated in the presence of 1 $\mu$M hydrolase for 1 h at 30°C. Reactions were stopped by addition of LDS sample buffer (Life Technologies) and incubation at 95°C for 3 min. Samples were then analyzed by SDS–PAGE and autoradiography.

### ARH3 expression and localization in patient fibroblasts

Fibroblasts were obtained from skin or cartilage biopsies from the probands of Family A and B. For all fibroblast experiments, two unrelated control fibroblast lines were used. Fibroblast cultures were maintained in DMEM high glucose (Thermo Fisher Scientific) with 10% FBS, 1% penicillin/streptomycin, and 1% L-glutamine and kept at 37°C and 5% $CO_2$.

Immunoblotting was performed on untreated fibroblastoid cell pellets and treated with RIPA lysate buffer (150 mM NaCl, 1% Triton

X-100, 0.5% sodium deoxycholate, 0.1% SDS, and 50 mM Tris–HCl [pH 8.0]), containing protease inhibitors (Roche) and phosphatase inhibitors (Roche). Protein concentration was measured by BCA assay kit (Invitrogen) according to the manufacturer's protocol. 4× NuPage LDS sample buffer (Thermo Fisher Scientific) was added to the lysates and samples were adjusted with RIPA lysate buffer to the same volume, after which they were boiled 95°C for 5 min before loading. 10 $\mu$g per sample was loaded onto a NuPAGE 4–12% gradient Bis-Tris SDS–PAGE gel (Thermo Fisher Scientific), 10 $\mu$l SeeBlue prestained ladder plus 2 (Thermo Fisher Scientific) was loaded as a size reference. Wet transfer was performed onto nitrocellulose blotting paper. Subsequent blocking was performed using 5% wt/vol milk-PBST. A primary mouse anti-ARH3 antibody (1 in 1,000) was used (sc-374162; Santa Cruz). A rabbit–anti-mouse IgG1 HRP-conjugated secondary antibody was used. Stripping and reprobing for GAPDH was performed to assess equal loading across all samples.

### Subcellular localization native and DNA damage

Two T175 flasks of fibroblasts were collected in DPBS per cell line (Gibco). The cells were washed twice in cold DPBS (Gibco). For subcellular fractionations, all buffers contained 2 $\mu$M of PARG and PARP inhibitor (PDD00017273 and olaparib). 100 $\mu$l of the cell suspension was used as whole-cell lysate fraction (wcl). For the wcl a cell pellet was created and resuspended in RIPA buffer and incubated on ice for 20 min and afterwards stored at −20°C.

The remaining 900 $\mu$l of cell suspension was centrifuged to create a cell pellet and subsequently resuspended in 700 $\mu$l fresh fractionation buffer (Hepes 6 mM, EGTA 0.125 mM, and 312 mM D-mannitol, adjusted to pH 7.5) and incubated on ice for 15 min. After incubation, the cell suspension was homogenized using a Micro lance size 26.5 G (BD) needle 10 times and incubated for another 10 min. Afterwards the suspension was centrifuged on 600 rcf for 10 min at 4°C. The obtained supernatant, containing the mitochondrial and cytosolic fraction, was transferred to a new 1.5-ml Eppendorf tube. The remaining pellet contained the nuclei and was left on ice until further processing.

The supernatant containing the mitochondrial and cytosolic fraction was centrifuged using 7,000$g$ for 10 min for at 4°C. The supernatant cytosolic fraction was transferred to a new 1.5-ml Eppendorf tube and placed in the freezer at −20°C. The remaining mitochondrial pellet was washed twice using 1 ml fractionation buffer and centrifuged at 7,000$g$ for 10 min at 4°C. The mitochondrial pellet was resuspended in 100 $\mu$l RIPA buffer for final lysis of the mitochondria. The suspension was incubated on ice for 20 min, sonicated, and stored at −20°C.

The nuclear pellet was washed twice using 1 ml fractionation buffer and centrifuged at 600$g$ for 10 min at 4°C. The nuclear pellet was resuspended in 300 $\mu$l fractionation buffer. During this step a Micro lance size 26.5 G (BD) needle was used to further homogenize the nuclear fraction and centrifuged at 600$g$ for 10 min at 4°C. The nuclear pellet was washed and spun down at 600$g$ for 10 min at 4°C. Then the nuclear pellet was resuspended in 100 $\mu$l RIPA buffer. The sample was incubated on ice for 20 min, sonicated, and stored at −20°C. Subsequent immunoblotting was performed as described

above for ARH3 and loading controls for the specific subcellular fractions α-tubulin (cytosolic), histone H3 (nuclear), lamin A/C (nuclear envelope), and VDAC1 (mitochondrial).

The same fractionation protocol used to analyze the subcellular localization of ARH3 in native conditions was used to assess it in DNA damage conditions. Although patient fibroblasts were subjected to 2 mM $H_2O_2$ for 10 min and then left to recuperate in fresh medium for 30 min before collection and subsequent fractionation.

### ARH3 live cell imaging and laser stripe recruitment

We used the standard human model cell lines U2OS (HTB-96; osteosarcoma; ATCC) and Be(2)-M17 (CRL-2267; neuroblastoma; ATCC). U2OS cells were grown in DMEM (Sigma-Aldrich) and Be(2)-M17 cells were grown in DMEM/F12 (Sigma-Aldrich) both supplemented with 10% FBS (Gibco) and penicillin–streptomycin (100 U/ml; Gibco) at 37°C with 5% $CO_2$. U2OS and Be(2)-M17 cells were seeded in 24 well glass bottomed cell culture dishes. The following day, cells were transfected with pDEST47-ARH3(WT)-GFP, pDEST47-ARH3(D77N/D78N)-GFP, or pDEST47-ARH3(V335G)-GFP for 24 h using TransIT-LT1 (Mirus) transfection reagent according to the manufacturers protocol. For laser stripe experiments, U2OS cells were incubated in complete DMEM supplemented with 10 μM BrdU (Sigma-Aldrich) for 24 h before imaging. On the day of imaging, transfected U2OS or Be(2)-M17 cells were then incubated with complete media supplemented with 100 nM MitoTracker Deep Red FM (Thermo Fisher Scientific) and 1 μg/ml Hoechst 33342 (Thermo Fisher Scientific) for 1 h at 37°C with 5% $CO_2$ to label mitochondria and nuclei, respectively. Laser stripe experiments were them performed as previously described (Gibbs-Seymour et al, 2016). Images of live cells were taken on the Olympus Fluoview FV1000 confocal microscope using a 60× oil objective.

## Supplementary Information

## Acknowledgements

This work was supported by the Association Belge contre les Maladies Neuromusculaire (ABMM)—Aide à la Recherche ASBL (2017-2018/05), the EU FP7/2007-2013 under grant agreement number 2012-305121 (NEURO-MICS), the EU Horizon 2020 program (Solve-RD under grant agreement No 779257), Work in I Ahel laboratory is supported by the Wellcome Trust (101794 and 210634), Biotechnology and Biological Sciences Research Council (BB/R007195/1), and Cancer Research United Kingdom (C35050/A22284). D Beijer is supported by a DOCPRO4 Antwerp University Research Fund (BOF) project grant under agreement number DOCPRO2016–33497. J Baets is supported by a Senior Clinical Researcher mandate of the Research Fund—Flanders (FWO) under grant agreement number 1805021N. Several authors of this publication are member of the European Reference Network for Rare Neuromuscular Diseases (ERN EURO-NMD). J Baets is is a member of the μNEURO Research Centre of Excellence of the University of Antwerp.

## Author Contributions

D Beijer: conceptualization, data curation, formal analysis, investigation, methodology, and writing—original draft, review, and editing.
T Agnew: conceptualization, data curation, formal analysis, investigation, methodology, and writing—original draft, review, and editing.
JGM Rack: data curation, formal analysis, investigation, and methodology.
E Prokhorova: investigation and methodology.
T Deconinck: investigation and methodology.
B Ceulemans: data curation and formal analysis.
S Peric: data curation and formal analysis.
V Milic Rasic: data curation and formal analysis.
P De Jonghe: data curation, formal analysis, supervision, and funding acquisition.
I Ahel: conceptualization, formal analysis, supervision, funding acquisition, methodology, and writing—original draft, review, and editing.
J Baets: conceptualization, supervision, funding acquisition, investigation, methodology, and writing—original draft, review, and editing.

## Conflict of Interest Statement

The authors declare that they have no conflict of interest.

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
