## [Reviewer comments · Life Science Alliance]

Life Science Alliance

Biallelic ADPRHL2 mutations in complex neuropathy affect ADP ribosylation and DNA damage response

Danique Beijer, Thomas Agnew, Johannes Rack, Evgeniia Prokhorova, Tine Deconinck, Berten Ceulemans, Stojan Peric, Vedrana Milic Rasic, Peter De Jonghe, Ivan Ahel, and Jonathan Baets
DOI: <https://doi.org/10.26508/lsa.202101057>

Corresponding author(s): Jonathan Baets, University of Antwerp and Ivan Ahel, University of Oxford

Review Timeline:

Submission Date:	2021-02-16
Editorial Decision:	2021-03-27
Revision Received:	2021-07-23
Editorial Decision:	2021-08-16
Revision Received:	2021-08-24
Accepted:	2021-08-25

Transaction Report:

March 27, 2021

Re: Life Science Alliance manuscript #LSA-2021-01057-T

Prof. Jonathan Baets
University of Antwerp
Universiteitsplein 1
Wilrijk, Antwerp 2610
Belgium

Dear Dr. Baets,

Thank you for submitting your manuscript entitled "BIALLELIC ADPRHL2 MUTATIONS IN COMPLEX NEUROPATHY AFFECT ADP RIBOSYLATION AND DNA DAMAGE RESPONSE" to Life Science Alliance. The manuscript was assessed by expert reviewers, whose comments are appended to this letter.

We apologize for this unusual and extended delay in getting back to you. As you will note from the reviewers' comments below, while both reviewers find the study interesting, they differ in their revision requests. We agree with all the points raised by both the reviewers, and would like to invite you to submit a revised version of this study that addresses all of the reviewers' concerns. In particular, the concern about subcellular localisation studies raised by Rev 2 must be addressed in the revision.

Thank you for this interesting contribution to Life Science Alliance. We are looking forward to receiving your revised manuscript.

Sincerely,

Shachi Bhatt, Ph.D.
Executive Editor
Life Science Alliance
<https://www.lsjournal.org/>
Tweet @SciBhatt @LSAJournal

- A letter addressing the reviewers' comments point by point.
- An editable version of the final text (.DOC or .DOCX) is needed for copyediting (no PDFs).
- High-resolution figure, supplementary figure and video files uploaded as individual files: See our detailed guidelines for preparing your production-ready images, <https://www.life-science-alliance.org/authors>
- Summary blurb (enter in submission system): A short text summarizing in a single sentence the study (max. 200 characters including spaces). This text is used in conjunction with the titles of papers, hence should be informative and complementary to the title and running title. It should describe the context and significance of the findings for a general readership; it should be written in the present tense and refer to the work in the third person. Author names should not be mentioned.

B. MANUSCRIPT ORGANIZATION AND FORMATTING:

Reviewer #1 (Comments to the Authors (Required)):

The study by Beijer et al. on the characterization of two mutations in ADPRHL2/ARH3 is a important contribution to the fields of both ADP-ribosylation and neurological disorders. Authors identify and characterize in detail two mutation in ARH3 identified in patients with hereditary motor neuropathy. The C26F mutation results in misfolding and consequently no detectable ARH3 in

patient fibroblasts carrying the mutation. The V335G mutation also leads to severe reduction in ARH3, while the mutant protein retains its enzymatic activity. Authors find that the V335G mutation specifically leads to the loss of a dramatic reduction in the nuclear pool of ARH3 with concomitant increase in nuclear ADP-ribosylation upon genotoxic stress. Thus, authors not only identify genetic further evidence for the role of ADP-ribosylation, in particular, ARH3 in neuronal function but they also identify a novel mode in the regulation of ARH3 activity that is its nuclear accumulation. The manuscript is well written and timely. The figures are easy to understand, well-described and of high quality. Authors comprehensively discuss their finding and its implications in the context of the current state-of-the-art.

Please refer to Figure 3c, d and e in the text.

Scale bar size missing on Fig 4. Scales bars are missing on Suppl. fig 2 a and b.

Reviewer #2 (Comments to the Authors (Required)):

Beijer & Agnew et al. report mutations in ADPRHL2 (ARH3) identified in two families. The mutations, V335G and C26F, are associated with neuropathy. The authors report clinical and genetic findings and characterise the protein products with regards to solubility, catalytic activity, subcellular localisation and their effects on poly(ADP-ribose) levels. V335G has previously been reported but not characterised to this extent.

The findings are biomedically relevant and interesting, particularly to researchers focussed on the DNA damage response, ADP-ribosylation and neurobiology.

With ARH3's function in removing serine ADP-ribosylation only recently reported, the study is timely.

I have a range of comments and concerns:

Main points

- p. 7, last paragraph: As glycine increases the flexibility of protein chains, the V335G mutation may render the affected part of the protein more mobile.
- The resolution of the structural representations (Figure 2) needs to be improved. The images appear very pixelated.
- Instead of stating the type of experiments in the Results sub-headings, the authors could state their key conclusions for the respective sections. This would be more meaningful and informative.
- p. 11, paragraph 1: The authors should perhaps point out that the V335G mutation does not appear to affect catalysis *in this particular assay*. It is an endpoint assay, and it of course remains unclear whether kinetics are affected by the mutation.
- The subcellular localisation studies require particular attention. First of all, it is unclear why the authors choose U2OS cells for microscopy studies when perhaps a neuronal cell line would have been more suitable. This becomes particularly important as subcellular localisation of the overexpressed protein appears to differ from that of the endogenous protein, as assessed by subcellular fractionation. In the last paragraph of p. 11, the authors state that the observations in

U2OS cells "somewhat differ[] from the cellular fractionation results in patient (fibroblast) cell lines". "Somewhat" is a strong understatement: the overexpressed ARH3(V335G)-GFP fusion appears predominantly cytoplasmic whilst the endogenous V335G mutant in patient fibroblasts is barely present in the cytoplasm (Figure 3c). This part of the study causes a lot of confusion. Does the biochemical fractionation protocol interfere with subcellular localisation? Is localisation cell-type specific? Can the ARH3 antibody be used in immunofluorescence microscopy to detect the endogenous protein? Would subcellular fractionation of cells overexpressing ARH3 give results that differ from microscopy? On page 19, the authors mention a potential piggyback mechanism for nuclear localisation of ARH3. Could the piggyback factor be cell-type specific? The choice of cell line should therefore be carefully considered and justified. It is important to resolve the confusion about ARH3 subcellular localisation, especially as the authors put a lot of emphasis on altered subcellular localisation as a potential disease mechanism (p. 20: "profound effect on nuclear localization"). The uncertainty remains unresolved and questions the conclusions made in the Discussion section.

- p. 12, last paragraph referring to Figure 4: There appears no specific mitochondrial localisation of ARH3. The cytoplasmic ARH3 signal is uniform and simply overlays with the mitotracker signal, but without enrichment at mitochondria. This of course raises questions about the "mitochondrial" fraction obtained biochemically (Figures 3 and 5). Isn't the "mitochondrial" fraction merely a general organellar fraction (see Methods)?

- p. 12, last lines: The microscopy studies merely consider the steady state. Hence, the authors cannot conclude that "nuclear import [is] [...] affected". (Equally likely, nuclear export may be promoted.)

- I am worried about Figure 4 and Supplementary Figure 2: The ARH3(V335G)-GFP panels in Supplementary Figure 2 are duplicated from the corresponding panels in Figure 4, just in a different orientation and a somewhat different field of view. What is the purpose of Supplementary Figure 2? It seems redundant with Figure 4, so I hope this is just a mistake. Also, micrographs in Supplementary Figure 2 are missing scale bars.

- p. 13, paragraph 1: I am missing the data involving olaparib (cross-reference is to Supplementary Figure 2).

- Supplementary Figure 2b: The cell expressing ARH3(V335G)-GFP does not display any nuclear signal for the protein, so it is not surprising to see a lack of enrichment in laser stripes. Was this experiment to co-evaluate the communication between the nuclear and cytoplasmic pools of the protein? As cells with substantial nuclear levels of ARH3(V335G)-GFP are shown in the paper, the authors could have selected such more appropriate cells to address their question. It is also advisable to quantify multiple experiments.

- p. 13, paragraph 2: The authors refer to experiments involving H₂O₂ treatment, but these are difficult to find. I eventually spotted the note in the legend for Figure 5a. Wouldn't it have been far more informative to perform the fractionations in the absence and presence of H₂O₂ in parallel and analyse them side by side? Separating this experiment in two parts adds to the confusion and uncertainty about ARH3 subcellular localisation.

- Micrograph figure legends do not state the size of the scale bar.

- Figures 3d, 3e, 5b, 5c: Statistical information is missing from the figure legends (n, mean?, nature of error bars)

- Figure 5c: PanADPr "expression" is not a particularly suitable term; I recommend "signal".
- Supplementary Figure 1: Is this an overlay of different images (marker vs. signal)?
- Supplementary Figure 3: What is the explanation for ARH3 "disappearing" upon H₂O₂ treatment?
- The methods for live cell imaging are incomplete; for example, mitotracker and Hoechst are not mentioned.
- The Methods section "Subcellular localization native and DNA damage" needs some attention. For example, entities and concentrations are written in reverse, and commas are used instead of decimals. Also, RCF should be expressed in multiples of g.
- On a very general note, how confident can the authors be that the disease phenotype can be ascribed to the ARH3 mutations?

Minor points

- p. 3, paragraph 1: "onto a target protein" - It will be more precise to say "target site" to avoid misunderstanding the stoichiometry of modification.
- p.4, beginning of paragraph 2: Pointing out that ADPRHL2 encodes ARH3 would be helpful.
- Punctuation (use of commas) can be improved throughout the manuscript. On page 5, a sentence starts with "whereas", which is stylistically suboptimal.
- Cell culture work is referred to as "in vivo", a term typically referring to model organisms, not cell lines.
- p. 13: Proteins "are recruited to" subcellular sites (not "recruit to").
- Figure 4: Label for ARH3(V335G)-GFP needs correction.

 I am worried about Figure 4 and Supplementary Figure 2: The ARH3(V335G)-GFP panels in Supplementary Figure 2 are duplicated from the corresponding panels in Figure 4, just in a different orientation and a somewhat different field of view. What is the purpose of Supplementary Figure 2? It seems redundant with Figure 4, so I hope this is just a mistake.

Reviewer #1

1. Please refer to Figure 3c, d and e in the text.

→ We have now included in-text reference to Figure 3c, d and e where appropriate.

2. Scale bar size missing on Fig 4. Scales bars are missing on Suppl. fig 2 a and b.

→ Supplementary figure 2a was removed due to a duplication with Figure 4. A new Supplementary figure 2a has been inserted with a scale bar included. The original Supplementary figure 2b now also has a scale bar.

Reviewer #2**Main points**

1. p. 7, last paragraph: As glycine increases the flexibility of protein chains, the V335G mutation may render the affected part of the protein more mobile.

→ We agree with the reviewer that the V335G mutation increases the flexibility of the loop it is situated in, which we argue could lead to the exposure of a hydrophobic pocket and protein destabilisation or degradation, e.g. via misfolded protein recognition. We consider this to be a likely pathway as we observe only trace amount of ARH3 V335G protein in patient cells and under conditions of overexpression. A more general increase of the flexibility of the protein region surrounding the Val335 position cannot be inferred from the structure as Val335 is packed against an alpha-helical bundle (which itself is deeply in-bedded in the core structure) with many hydrophobic interaction at its core, thus it appears unlikely to us that removal of an external interaction decreases the stability of this structural element. On its sides the Val335 containing loop borders the N-terminal region and the Glu41-flap, respectively. These two regions are already in the wt protein very flexible and often unresolved in available crystal structures, thus we would speculate that no further increase in flexibility can be transferred into these regions due to the V335G mutation.

2. The resolution of the structural representations (Figure 2) needs to be improved. The images appear very pixelated.

→ The low resolution of the images in Figure 2 was likely due to compression at some stage of the submission. We have now made sure to submit high resolution images as separate files for all the figures to solve this issue.

3. Instead of stating the type of experiments in the Results sub-headings, the authors could state their key conclusions for the respective sections. This would be more meaningful and informative.

→ We have changed the section sub-headers for the Results section to describe the key findings of the section.

4. p. 11, paragraph 1: The authors should perhaps point out that the V335G mutation does not appear to affect catalysis *in this particular assay*. It is an endpoint assay, and it of course remains unclear whether kinetics are affected by the mutation.

→ We agree with the reviewer that there are limitations to our assay and as such we have stated

more clearly in the text that the V335G mutation doesn't affect enzymatic activity in this particular endpoint assay.

5. The subcellular localisation studies require particular attention. First of all, it is unclear why the authors choose U2OS cells for microscopy studies when perhaps a neuronal cell line would have been more suitable. This becomes particularly important as subcellular localisation of the overexpressed protein appears to differ from that of the endogenous protein, as assessed by subcellular fractionation. In the last paragraph of p. 11, the authors state that the observations in U2OS cells "somewhat differ from the cellular fractionation results in patient (fibroblast) cell lines". "Somewhat" is a strong understatement: the overexpressed ARH3(V335G)-GFP fusion appears predominantly cytoplasmic whilst the endogenous V335G mutant in patient fibroblasts is barely present in the cytoplasm (Figure 3c). This part of the study causes a lot of confusion. Does the biochemical fractionation protocol interfere with subcellular localisation? Is localisation cell-type specific? Can the ARH3 antibody be used in immunofluorescence microscopy to detect the endogenous protein? Would subcellular fractionation of cells overexpressing ARH3 give results that differ from microscopy? On page 19, the authors mention a potential piggyback mechanism for nuclear localisation of ARH3. Could the piggyback factor be cell-type specific? The choice of cell line should therefore be carefully considered and justified. It is important to resolve the confusion about ARH3 subcellular localisation, especially as the authors put a lot of emphasis on altered subcellular localisation as a potential disease mechanism (p. 20: "profound effect on nuclear localization"). The uncertainty remains unresolved and questions the conclusions made in the Discussion section.

→ We agree with the reviewer that it is possible that differences in subcellular localization of ARH3 may be cell type specific. Therefore we repeated our over-expression analysis of WT and mutant ARH3-GFP protein in a neuroblastoma cell line (Be(2)M17) as it might be more relevant to neuronal phenotypes and include this data in our revised manuscript as a new figure (Figure S2A). The results using a neuronal cell line mirrored those from U2OS cells showing loss of nuclear and retention of mitochondrial ARH3(V335G)-GFP protein. The inclusion of these neuronal cell line data that corroborate our initial findings show that these observations are not restricted to U2OS cells and that regulation of ARH3 sub-cellular localization is not cell line dependent, strengthening our conclusions that loss of nuclear ARH3 protein/function likely plays a role in the pathogenesis of patients.

As stated in our original manuscript, the antibody that we used for WB is not suitable for immunofluorescent analysis. We therefore sought to validate our fractionation data via an alternative method choosing to perform live-cell imaging of over-expressed WT and mutant proteins in U2OS cells confirming that mitochondrial localization of ARH3-V335G is unaffected but that nuclear translocation is affected. We can make these conclusions because we have included sufficient controls. Both WT and catalytically dead ARH3 protein both localize to the cytoplasm, nucleus and mitochondria when over-expressed however the V335G mutant protein does not localize to the nucleus. The conclusion from this being that the V335G mutation affects nuclear import/export. Previously it has been shown that MACROD2, another ADPr hydrolase, is exported from the nucleus following DNA damage due to phosphorylation by ATM to sequester MACROD2 from DNA damage sites (Golia et al., 2017). It is possible therefore that ARH3 sub-cellular localization is regulated in a similar manner that is reliant of Valine335. As the reviewer rightly states, fractionation data showed reduced cytoplasmic endogenous ARH3-V335G protein whilst the over-expressed ARH3-V335G protein was mainly cytoplasmic. This is explained as the over-expressed protein is under the control of a strong promoter resulting in a large quantities of mutant protein being synthesized in the cytoplasm before being trafficked around the cell. The rate of synthesis of mutant ARH3-V335G likely exceeds the rate of degradation in our over-expressed system whereas this is likely not the case for endogenous protein hence the difference between the two methodologies.

The main conclusion from our data is that the V335G mutation has given us the unique opportunity of separating the mitochondrial and nuclear / cytoplasmic functions of ARH3. Our discovery that the mitochondrial pool of ARH3 is unaffected in V335G patients and OE models, allowed us to focus on the nuclear function of ARH3 in DNA damage response as a cause of the neurodegenerative phenotype. Indeed, this is in line with our recent observation that loss of ARH3 causes chromatin dysregulation and stress-induced PARP1 dependent cell death due to the accumulation of unrestrained nuclear ADP-ribosylation (Prokhorova et al; Molecular Cell, 2021) which we believe is likely the mechanism of neurodegeneration in patients with ARH3 mutations.

6. p. 12, last paragraph referring to Figure 4: There appears no specific mitochondrial localisation of ARH3. The cytoplasmic ARH3 signal is uniform and simply overlays with the mitotracker signal, but without enrichment at mitochondria. This of course raises questions about the "mitochondrial" fraction obtained biochemically (Figures 3 and 5). Isn't the "mitochondrial" fraction merely a general organellar fraction (see Methods)?

→ The mitochondrial localization of ARH3 has previously been established (Mashimo, Kato, & Moss, 2013; Niere, Kernstock, Koch-Nolte, & Ziegler, 2008; Niere et al., 2012) and our data from the overexpression model and the patient-derived model both support this. The live-cell imaging of ARH3-GFP in U2OS cells clearly shows overlapping signals of mitotracker (red) and ARH3 (green) (Figure 4) indicative of mitochondrial localization, this may be more clearly seen now that we have increased the resolution of our images. The subcellular fractionation experiment similarly supports this as shown by the high VDAC1 expression in the mitochondrial fraction, demonstrating a fraction clearly enriched for mitochondria and indeed presence of ARH3 in this mitochondria enriched fraction again consistent with previous findings.

7. p. 12, last lines: The microscopy studies merely consider the steady state. Hence, the authors cannot conclude that "nuclear import [is] [...] affected". (Equally likely, nuclear export may be promoted.)

→ We have adjusted this sentence to indeed reflect the limitations of our experiments in assessing differences between import and export.

8. I am worried about Figure 4 and Supplementary Figure 2: The ARH3(V335G)-GFP panels in Supplementary Figure 2 are duplicated from the corresponding panels in Figure 4, just in a different orientation and a somewhat different field of view. What is the purpose of Supplementary Figure 2? It seems redundant with Figure 4, so I hope this is just a mistake. Also, micrographs in Supplementary Figure 2 are missing scale bars.

→ We apologize for the mistake with this figure which was an oversight on our part, when we decided to move Supplementary figure 2 panel A to the main text as Figure 4 but forgot to remove it from the supplementary figure as well. This is indicated by the identical figure legend used for both. We have now removed the superfluous Supplementary figure 2 panel A.

9. p. 13, paragraph 1: I am missing the data involving olaparib (cross-reference is to Supplementary Figure 2).

→ We were considering including a supplementary figure showing that ARH3 recruits to laser stripe

induced DNA damage sites in an PARP dependent manner (this would have been supplementary figure 2A). However, considering that this finding has already been previously published (Wang et al., 2018), we decided that it was unnecessary to include this data in our paper. We will amend the main text to correctly reference this finding from Wang *et al.* and will completely remove supplementary figure 2A which was included by mistake.

10. Supplementary Figure 2b: The cell expressing ARH3(V335G)-GFP does not display any nuclear signal for the protein, so it is not surprising to see a lack of enrichment in laser stripes. Was this experiment to co-evaluate the communication between the nuclear and cytoplasmic pools of the protein? As cells with substantial nuclear levels of ARH3(V335G)-GFP are shown in the paper, the authors could have selected such more appropriate cells to address their question. It is also advisable to quantify multiple experiments.

→ There are several examples in the literature of proteins that translocate from the cytoplasm to the nucleus upon DNA damage eg. APLF (Mehrotra et al., 2011). More notably, a ADP-ribosyl hydrolase enzyme MACROD2 (Golia et al., 2017), nucleo/cytoplasmic shuttling is regulated by the DNA damage response kinase ATM where MACROD2 is phosphorylated and exported from the nucleus following DNA damage into the cytosol. The laser stripe assays performed were to establish whether ARH3-V335G can translocate from the cytoplasm to the nucleus upon genotoxic stress. The experiment clearly shows the V335G ARH3 protein is not imported into the nucleus from the cytosol following DNA damage supporting our findings that there is reduced nuclear function of ARH3 in ARH3-V335G patient cells. We have amended the results section for these data in the main text to better outline the aim of these experiments.

11. p. 13, paragraph 2: The authors refer to experiments involving H₂O₂ treatment, but these are difficult to find. I eventually spotted the note in the legend for Figure 5a. Wouldn't it have been far more informative to perform the fractionations in the absence and presence of H₂O₂ in parallel and analyse them side by side? Separating this experiment in two parts adds to the confusion and uncertainty about ARH3 subcellular localisation.

→ We have aimed to improve the cross referencing by altering the text and by referring to the relevant panels in both figures 3 and 5. While we agree that a side-by-side analysis of H₂O₂ stimulated and non-stimulated cells would have been interesting, there are practical limitations to the growing of sufficient patient-derived cells for all conditions simultaneously as well as the processing of these cells into cell lysates in the same experiment. We feel as if the current separation was least harmful to the overall interpretation of the subcellular localization experiments as it was most interesting to see difference between the wild-type and the V335G mutant within each experimental condition.

12. Micrograph figure legends do not state the size of the scale bar.

→ Figure 4 and Supplementary figure 2 now have the size of the scale bar indicated in the figure legend.

13. Figures 3d, 3e, 5b, 5c: Statistical information is missing from the figure legends (n, mean?, nature of error bars)

→ We have now added the statistical information for the tests used regarding the subcellular localization quantification experiments, which included n=4, mean and SD for all experiments.

14 Figure 5c: PanADPr "expression" is not a particularly suitable term; I recommend "signal".

→ We have adjusted the labelling in both Figure 5c as well as Figure 3e.

15. Supplementary Figure 1: Is this an overlay of different images (marker vs. signal)?

→ Yes, this is the overlay as generated by the imaging device itself (marker vs. signal), no manual overlay was used here.

16. Supplementary Figure 3: What is the explanation for ARH3 "disappearing" upon H2O2 treatment?

→ This is indeed an interesting observation that we made too and seems to be consistent throughout several models that we have used. ARH3 levels seem to be lower after H2O2 stimulation in both the control (cntr) and V335G fibroblast cells lines (Figure S3). We speculate that upon H2O2 stimulation and recruitment to DNA damage sites, ARH3 is cleaved / degraded in a regulated manner, the function of which remains unclear and is beyond the scope of this study. Given this effect is observed for both wild-type ARH3 and V335G ARH3 we believe that this does not play a role in the proposed mechanism of the V335G mutant pathology.

17. The methods for live cell imaging are incomplete; for example, mitotracker and Hoechst are not mentioned.

→ The methods section for the live cell imaging has been updated.

18. The Methods section "Subcellular localization native and DNA damage" needs some attention. For example, entities and concentrations are written in reverse, and commas are used instead of decimals. Also, RCF should be expressed in multiples of g.

→ We have adapted the centrifuge speed to times g and used decimals instead where applicable.

19. On a very general note, how confident can the authors be that the disease phenotype can be ascribed to the ARH3 mutations?

→ In family A, both unaffected parents and the affected siblings were subjected to whole exome sequencing. In family B, both the proband and the unaffected father were subjected to whole

genome sequencing. We first excluded variants in known neurological disease genes associated with the phenotypes in the specific patients either based on the initial filtering criteria or subsequently by lack of segregation in the family. Subsequently, we assessed other genetic variants and based on the high pathogenic prediction scores and the segregation with the phenotype the variants in *ADPRHL2* are the most likely causal variants in each of the families. The V335G mutation present in Family A has also been published in two other families since and the C26F with severe loss of protein expression fit well with several reports of truncating *ADPRHL2* mutation causing similar phenotypes. This means that both families were studied with state-of-the-art genetic techniques that provided unbiased exome- and genome-wide data producing the *ADPRHL2* mutations as the highly likely cause of disease based on stringent filtering and segregation analysis.

Minor points

20. p. 3, paragraph 1: "onto a target protein" - It will be more precise to say "target site" to avoid misunderstanding the stoichiometry of modification.

→ We have adjusted this sentence accordingly.

21. p.4, beginning of paragraph 2: Pointing out that *ADPRHL2* encodes ARH3 would be helpful.

→ We have now clarified this in the text.

22. Punctuation (use of commas) can be improved throughout the manuscript. On page 5, a sentence starts with "whereas", which is stylistically suboptimal.

→ We have aimed to improve the overall style of the manuscript including use of punctuation.

23. Cell culture work is referred to as "in vivo", a term typically referring to model organisms, not cell lines.

→ We have replaced the references to *in vivo* work when referring to patient-derived cells.

24. p. 13: Proteins "are recruited to" subcellular sites (not "recruit to").

→ We have corrected multiple instances of this now, thank you.

25. Figure 4: Label for ARH3(V335G)-GFP needs correction.

→ We have corrected the label in figure 4.

References

- Golia, B., Moeller, G. K., Jankevicius, G., Schmidt, A., Hegele, A., Preisser, J., . . . Timinszky, G. (2017). ATM induces MacroD2 nuclear export upon DNA damage. *Nucleic Acids Res*, *45*(1), 244-254. doi:10.1093/nar/gkw904
- Mashimo, M., Kato, J., & Moss, J. (2013). ADP-ribosyl-acceptor hydrolase 3 regulates poly (ADP-ribose) degradation and cell death during oxidative stress. *Proc Natl Acad Sci U S A*, *110*(47), 18964-18969. doi:10.1073/pnas.1312783110
- Mehrotra, P., Riley, J. P., Patel, R., Li, F., Voss, L., & Goenka, S. (2011). PARP-14 functions as a transcriptional switch for Stat6-dependent gene activation. *J Biol Chem*, *286*(3), 1767-1776. doi:10.1074/jbc.M110.157768
- Niere, M., Kernstock, S., Koch-Nolte, F., & Ziegler, M. (2008). Functional localization of two poly(ADP-ribose)-degrading enzymes to the mitochondrial matrix. *Mol Cell Biol*, *28*(2), 814-824. doi:10.1128/MCB.01766-07
- Niere, M., Mashimo, M., Agledal, L., Dolle, C., Kasamatsu, A., Kato, J., . . . Ziegler, M. (2012). ADP-ribosylhydrolase 3 (ARH3), not poly(ADP-ribose) glycohydrolase (PARG) isoforms, is responsible for degradation of mitochondrial matrix-associated poly(ADP-ribose). *J Biol Chem*, *287*(20), 16088-16102. doi:10.1074/jbc.M112.349183
- Wang, M., Yuan, Z., Xie, R., Ma, Y., Liu, X., & Yu, X. (2018). Structure-function analyses reveal the mechanism of the ARH3-dependent hydrolysis of ADP-ribosylation. *J Biol Chem*, *293*(37), 14470-14480. doi:10.1074/jbc.RA118.004284

August 16, 2021

RE: Life Science Alliance Manuscript #LSA-2021-01057-TR

Prof. Jonathan Baets
University of Antwerp
Universiteitsplein 1
Wilrijk, Antwerp 2610
Belgium

Dear Dr. Baets,

Thank you for submitting your revised manuscript entitled "Biallelic ADPRHL2 mutations in complex neuropathy affect ADP ribosylation and DNA damage response". We would be happy to publish your paper in Life Science Alliance pending final revisions necessary to meet our formatting guidelines. Please also address Reviewer 2's remaining comments.

- please add a Category for your manuscript in our system
- please use the [10 author names, et al.] format in your references (i.e. limit the author names to the first 10)
- please add the Twitter handle of your host institute/organization as well as your own or one of the first author in our system
- please add your main, supplementary figure, and table legends to the main manuscript text after the references section
- please add callouts for Figures 5B and S2B-C to your main manuscript text
- for the work using patient samples, you mention informed consent, but please also include details on the study authorization
- please add size markers next to the blot in Figure S1

LSA now encourages authors to provide a 30-60 second video where the study is briefly explained. We will use these videos on social media to promote the published paper and the presenting author. Corresponding or first-authors are welcome to submit the video. Please submit only one video per manuscript. The video can be emailed to contact@life-science-alliance.org

A. FINAL FILES:

B. MANUSCRIPT ORGANIZATION AND FORMATTING:

Sincerely,

Reviewer #2 (Comments to the Authors (Required)):

I thank the authors for their revision! Whilst this improves the manuscript, some further work (solely on the level of manuscript editing) is, in my eyes, minimally required before publication. It is somewhat unfortunate that the apparently contradictory subcellular localisation data still leave questions - more controls could have been performed, but this is of course up to the authors. Nonetheless, I feel that these open questions should at least be partially addressed by a frank discussion of the data.

- abstract: "modification mediated by poly(ADP-ribose)polymerases" - I recommend using the more general term "ADP-ribosyltransferases (ARTs)", which encompass both mono- and poly-ARTs.

- In general, can the authors adopt the new nomenclature for ADP-ribosyltransferases throughout (see Luescher et al., 2021)?

- p. 9: "... overexpression models using osteosarcoma U2OS and neuroblastoma BE(2)-M17 cells somewhat differs from the cellular fractionation results" - As stated previously, the term "somewhat" does definitely not accurately describe the data. The results from fractionation and immunofluorescence are markedly different. The added fluorescence microscopy data in neuroblastoma cells are helpful. Nonetheless, the reader is left puzzled as to whether the localisation differences between endogenous and overexpressed protein are dependent on the cell type, the overexpression construct (e.g., a potentially interfering GFP tag), or the technique used to assess subcellular localisation, among other possibilities. As these control experiments are straightforward to perform, I am surprised the authors did not address this point more extensively. Minimally, the localisation differences should be discussed openly and not understated, as is currently the case. The rebuttal letter contains part of this discussion, which could be covered in the manuscript itself.

- Mitochondrial localisation of the overexpressed protein still cannot be sufficiently appreciated from the fluorescence microscopy images. Could an inset showing an appropriately magnified area help? With clear mitochondrial localisation, you would expect the mitochondrial pattern to be recapitulated in the GFP channel.

- The reduction in ARH3 levels after H₂O₂ treatment should be mentioned in the text, and also discussed. This observation is, as the authors state in the rebuttal, indeed interesting.

- p. 10: "Wild-type control cells clearly show a loss of cytosolic ARH3 and an increase in nuclear ARH3 following H₂O₂ treatment indicating that ARH3 is imported from the cytosol following DNA damage (Fig 5 and Fig S3)" and subsequently data described in this section - The section describes changes in subcellular localisation in response to H₂O₂ treatment. However, the data from Fig. 5 and S3 do not include a minus-H₂O₂ condition, so the comparison this section refers to has actually not been made. The same comment applies to the Discussion section. This requires rewording: the comparison can be made between wild-type and mutant ARH3, not between

untreated and treated.

- p. 14, Discussion: "The V335G mutant shows somewhat reduced expression levels and normal (ADP- ribosyl)hydrolase activity, but altered subcellular localization of the ARH3 protein, with a reduced localization to the nucleus and cytosol." - I suggest the following edit: "... with a reduced localization, dependent on experimental system or possibly cell type, to either the nucleus or cytosol, with unaltered mitochondrial targeting."
- p. 5: "Whereas the V335G mutation results ..."  "Conversely, the V335G ..."
- p. 5: "Both patients in family A, were ..." - The comma is not necessary. There are numerous other instances of unnecessary commas, or missing commas, in the manuscript.
- p. 9: "... and fails to recruit to following pre-treatment with the PARP1/2 inhibitor olaparib ..." - this part of the sentence is unclear.
- p.9/10: The laser irradiation experiment lacks a crossreference to the figure (S2).
- The copyeditor can address stylistic matters, which include punctuation. Readability would improve.

Comment to the journal: Please ask authors to use page numbers, ideally also line numbers. Thank you!

August 25, 2021

RE: Life Science Alliance Manuscript #LSA-2021-01057-TRR

Prof. Jonathan Baets
University of Antwerp
Universiteitsplein 1
Wilrijk, Antwerp 2610
Belgium

Dear Dr. Baets,

Thank you for submitting your Research Article entitled "Biallelic ADPRHL2 mutations in complex neuropathy affect ADP ribosylation and DNA damage response". It is a pleasure to let you know that your manuscript is now accepted for publication in Life Science Alliance. Congratulations on this interesting work.

DISTRIBUTION OF MATERIALS:

Again, congratulations on a very nice paper. I hope you found the review process to be constructive and are pleased with how the manuscript was handled editorially. We look forward to future exciting submissions from your lab.

Sincerely,
